# Reconstruction of natural images from responses of primate retinal ganglion cells

Nora Brackbill[1]*, Colleen Rhoades[2], Alexandra Kling[3,4,5], Nishal P Shah[6], Alexander Sher[7], Alan M Litke[7], EJ Chichilnisky[3,4,5]

[1]Department of Physics, Stanford University, Stanford, United States; [2]Department of Bioengineering, Stanford University, Stanford, United States; [3]Department of Neurosurgery, Stanford School of Medicine, Stanford, United States; [4]Department of Ophthalmology, Stanford University, Stanford, United States; [5]Hansen Experimental Physics Laboratory, Stanford University, Stanford, United States; [6]Department of Electrical Engineering, Stanford University, Stanford, United States; [7]Santa Cruz Institute for Particle Physics, University of California, Santa Cruz, Santa Cruz, United States

**Abstract** The visual message conveyed by a retinal ganglion cell (RGC) is often summarized by its spatial receptive field, but in principle also depends on the responses of other RGCs and natural image statistics. This possibility was explored by linear reconstruction of natural images from responses of the four numerically-dominant macaque RGC types. Reconstructions were highly consistent across retinas. The optimal reconstruction filter for each RGC – its visual message – reflected natural image statistics, and resembled the receptive field only when nearby, same-type cells were included. ON and OFF cells conveyed largely independent, complementary representations, and parasol and midget cells conveyed distinct features. Correlated activity and nonlinearities had statistically significant but minor effects on reconstruction. Simulated reconstructions, using linear-nonlinear cascade models of RGC light responses that incorporated measured spatial properties and nonlinearities, produced similar results. Spatiotemporal reconstructions exhibited similar spatial properties, suggesting that the results are relevant for natural vision.

**\*For correspondence:**
nbrackbill@gmail.com

**Competing interests:** The authors declare that no competing interests exist.

## Introduction

The brain uses visual information transmitted by retinal neurons to make inferences about the external world. Traditionally, the visual signal transmitted by an individual retinal ganglion cell (RGC) has been summarized by its spatial profile of light sensitivity, or receptive field (RF), measured with stimuli such as spots or bars (*Chichilnisky, 2001*; *Kuffler, 1953*; *Lettvin et al., 1959*). Although intuitively appealing, this description may not reveal how the spikes from a RGC contribute to the visual representation in the brain under natural viewing conditions. In particular, because of the strong spatial correlations in natural images (*Ruderman and Bialek, 1994*), the response of a single RGC contains information about visual space well beyond its RF. Thus, across the RGC population, the responses of many individual cells could contain information about the same region of visual space, and it is not obvious how the brain could exploit this potentially redundant information (*Puchalla et al., 2005*). Complicating this issue is the fact that there are roughly twenty RGC types, each covering all of visual space with their RFs, and each with different spatial, temporal, and chromatic sensitivity (*Dacey et al., 2003*). Furthermore, RGCs show both stimulus-induced and stimulus-independent correlated activity, within and across cell types (*Greschner et al., 2011*; *Mastronarde, 1983*), which could substantially influence the encoding of the stimulus (*Meytlis et al., 2012*;

**eLife digest** Vision begins in the retina, the layer of tissue that lines the back of the eye. Light-sensitive cells called rods and cones absorb incoming light and convert it into electrical signals. They pass these signals to neurons called retinal ganglion cells (RGCs), which convert them into electrical signals called spikes. Spikes from RGCs then travel along the optic nerve to the brain. They are the only source of visual information that the brain receives. From this information, the brain constructs our entire visual world.

The primate retina contains roughly 20 types of RGCs. Each encodes a different visual feature, such as the presence of bright spots of a certain size, or information about texture and movement. But exactly what input each RGC sends to the brain, and how the brain uses this information, is unclear. Brackbill et al. set out to answer these questions by measuring and analyzing the electrical activity in isolated retinas from macaque monkeys. Studying the macaque retina was important because the primate visual system differs from that of other species in several ways. These include the numbers and types of RGCs present in the retina. These primates are also similar to humans in their high-resolution central vision and trichromatic color vision.

Using electrode arrays to monitor hundreds of RGCs at the same time, Brackbill et al. recorded the responses of macaque retinas to real-life images of landscapes, objects, animals or people. Based on these recordings, plus existing knowledge about RGC responses, Brackbill et al. then attempted to reconstruct the original images using just the electrical activity recorded. The resulting reconstructions were similar across all retinas tested. Moreover, they showed a striking resemblance to the original images. These results made it possible to comprehend how the light-response properties of each cell represent visual information that can be used by the brain.

Understanding how macaque retinas work in natural conditions is critical to decoding how our own retinas process and convey information. A better knowledge of how the brain uses this input to generate images could ultimately make it possible to design artificial retinas to restore vision in patients with certain forms of blindness.

*Pillow et al., 2008*; *Ruda et al., 2020*; *Zylberberg et al., 2016*). For these reasons, the visual message transmitted by a RGC to the brain is not fully understood.

One way to understand how each RGC contributes to vision is to determine how a natural image can be reconstructed from the light-evoked responses of the entire RGC population. This analysis approach mimics the challenge faced by the brain: using sensory inputs to make inferences about the visual environment (*Bialek et al., 1991*; *Rieke et al., 1997*). In the simplest case of linear reconstruction, the visual message of an individual RGC can be summarized by its optimal reconstruction filter, that is its contribution to the reconstructed image. Linear reconstruction has been used to estimate the temporal structure of a spatially uniform stimulus from the responses of salamander RGCs, revealing that reconstruction filters varied widely and depended heavily on the other RGCs included in the reconstruction (*Warland et al., 1997*). However, no spatial information was explored, and only a small number of RGCs of unknown types were examined. A later study linearly reconstructed spatiotemporal natural movies from the activity of neurons in the cat LGN (*Stanley et al., 1999*). However, neurons from many recordings were pooled, without cell type identification or the systematic spatial organization expected from complete populations of multiple cell types. More recently, several studies have used nonlinear and machine learning methods for reconstruction (*Botella-Soler et al., 2018*; *Parthasarathy et al., 2017*; *Zhang et al., 2020*), although these techniques were not tested in primate, or on large-scale data sets with clear cell type identifications and complete populations of RGCs. Thus, it remains unclear what spatial visual message primate RGCs convey to the brain, in the context of natural scenes and the full neural population.

We performed linear reconstruction of flashed natural images from the responses of hundreds of RGCs in macaque retina, using large-scale, multi-electrode recordings. These recordings provided simultaneous access to the visual signals of nearly complete populations of ON and OFF parasol cells, as well as locally complete populations of ON and OFF midget cells, the four numerically dominant RGC types that provide high-resolution visual information to the brain (*Dacey et al., 2003*). Data from 15 recordings produced strikingly similar reconstructions. Examination of reconstruction

filters revealed that the visual message of a given RGC depended on the responses of other RGCs, due to the statistics of natural scenes. Reconstruction from complete cell type populations revealed that they conveyed different features of the visual scene, consistent with their distinct light response properties. The spatial information carried by one type was mostly unaffected by the contributions of other types, particularly types with the opposite response polarity (ON vs. OFF). Two simple tests of nonlinear reconstruction revealed only minor improvements over linear reconstruction. Similar visual messages and reconstructions were obtained using linear-nonlinear cascade models of RGC light response incorporating measured spatial properties and response nonlinearity. Finally, full spatiotemporal reconstruction with dynamic scenes revealed similar spatial visual messages, suggesting that these findings may generalize to natural vision.

## Results

Large-scale multi-electrode recordings from the peripheral macaque retina were used to characterize light responses in complete populations of RGCs (*Chichilnisky and Kalmar, 2002*; *Field et al., 2010*; *Frechette et al., 2005*; *Litke et al., 2004*). The classical RF of each cell was measured by reverse correlation between its spike train and a spatiotemporal white noise stimulus, resulting in a spike-triggered average (STA) stimulus that summarized the spatial, temporal and chromatic properties of the cell (*Chichilnisky, 2001*). Clustering of these properties revealed multiple identifiable and complete cell type populations (see Materials and methods; *Chichilnisky and Kalmar, 2002*; *Dacey, 1993*; *Devries and Baylor, 1997*; *Field et al., 2007*; *Frechette et al., 2005*), including the four numerically dominant RGC types in macaque: ON parasol, OFF parasol, ON midget, and OFF midget. The RFs of each identified type formed an orderly lattice (mosaic), consistent with the spatial organization of each RGC type known from anatomical studies (*Wässle et al., 1983*).

Responses to natural images were then characterized by displaying static, grayscale images from the ImageNet database, which contains a wide variety of subjects including landscapes, objects, people, and animals (*Fei-Fei et al., 2009*). Each image was displayed for 100 ms, separated by 400 ms of spatially uniform illumination with intensity equal to the mean intensity across all images (*Figure 1A*). This stimulus timing produced a strong initial response from both parasol and midget cells, and a return to maintained firing rates prior to the onset of the next image. For each image, the population response was quantified as a vector of RGC spike counts in the 150 ms window after image onset (*Figure 1B*; window chosen to optimize reconstruction performance; see Materials and methods). The stimulus (S, dimensions: number of images x number of pixels) was reconstructed from the recorded ON and OFF parasol and midget cell responses (R, dimensions: number of images x number of cells) using a linear model, S = RW. The optimal weights for the linear model (W, dimensions: number of cells x number of pixels) were calculated using least squares regression,

$$W_{ls} = (R^T R)^{-1} R^T S. \tag{1}$$

The weights were then used to reconstruct a held-out set of test images. Reconstruction performance was measured by comparing only the areas of the original and reconstructed images covered by the RF mosaic for each RGC type included in the analysis (see Materials and methods). Pearson's linear correlation coefficient ($\rho$) was used as the performance metric; mean squared error (MSE) and the structural similarity (SSIM; *Wang et al., 2004*) showed the same trends. All statistical tests were computed using resampling to generate null models (see Materials and methods). Regularization of reconstruction weights was not necessary, because the number of samples was much larger (>20 x) than the number of parameters in all cases (see Materials and methods). In what follows, reconstruction 'from RGCs' is used as a shorthand to indicate reconstruction from their recorded responses, as described above.

The basic characteristics of spatial linear reconstruction were evaluated by reconstructing images from the responses of populations of ON and OFF parasol cells in 15 recordings from nine monkeys. In each case, both cell types formed complete or nearly complete mosaics with uniform coverage, indicating that nearly every cell of each type over the electrode array was recorded (see *Figures 1C* and *2*). Thus, the reconstructions revealed the full visual representation in these RGC populations. In each recording, reconstruction performance varied considerably across the set of test images

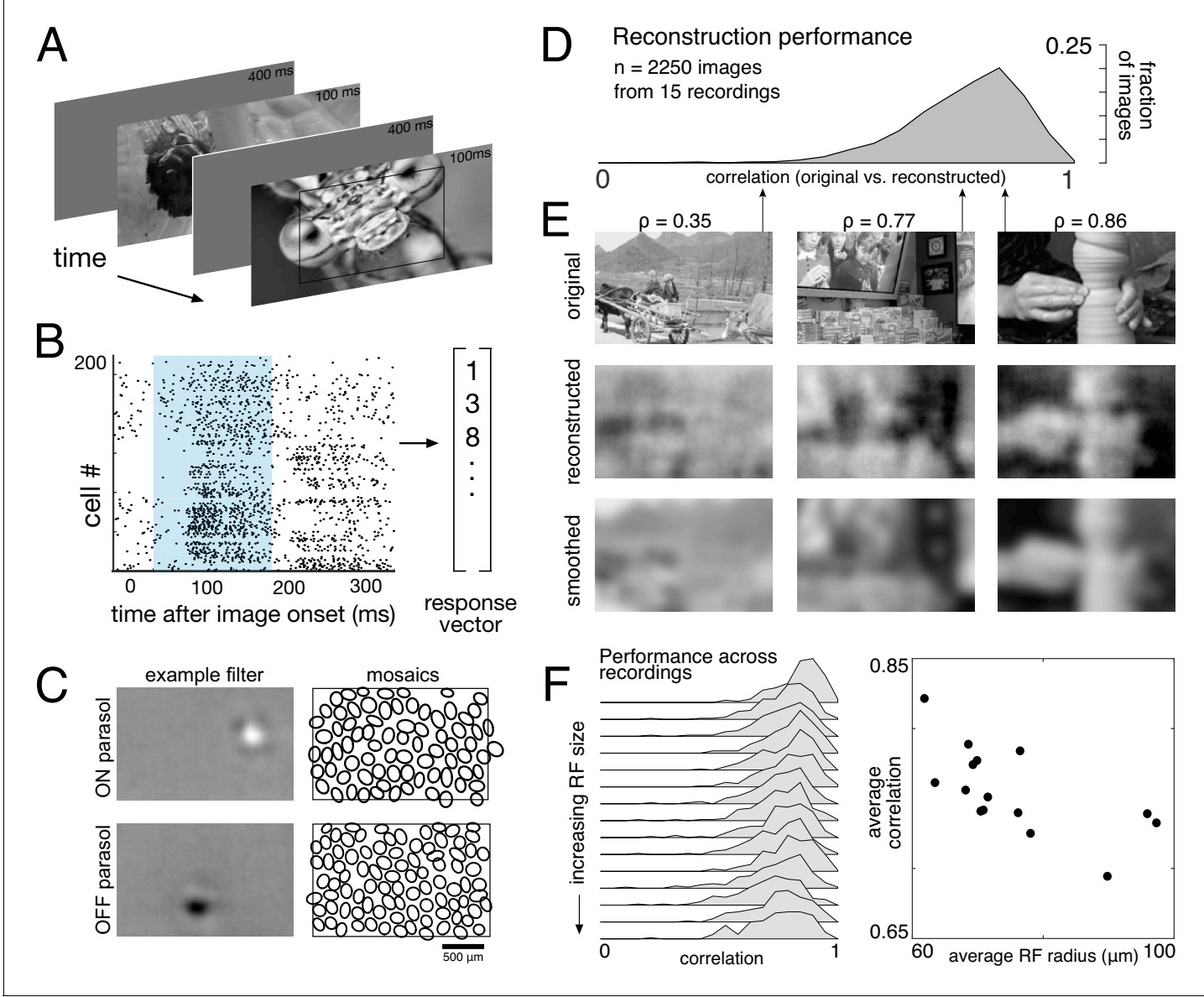

**Figure 1.** Linear reconstruction from ON and OFF parasol cell responses. (A) Visual stimulus: static images from the ImageNet database were displayed for 100 ms, with 400 ms of gray between. The thin black rectangle indicates the central image region shown in C and E. (B) Example population response: each entry corresponds to the number of spikes from one RGC in a 150 ms window (shown in blue) after the image onset. (C) Left: Examples of reconstruction filters for an ON (top) and OFF (bottom) parasol cell. Right: RF locations for the entire population of ON (top) and OFF (bottom) parasol cells used in one recording. (D) Reconstruction performance (correlation) across all recordings. (E) Example reconstructions for three representative scores (middle row), compared to original images (top row) and smoothed images (bottom row), from the same recording and at the same scale as shown in C. (F) Reconstruction performance across 15 recordings. Left: Distributions of scores across images for each recording, ordered by average receptive field (RF) size. Right: Average reconstruction performance vs. average RF radius ($\rho = -0.7$). Source files for D and F are available in *Figure 1—source data 1*.

The online version of this article includes the following source data for figure 1:

**Source data 1.** Linear reconstruction from ON and OFF parasol cell responses.

(*Figure 1D*, $\rho = 0.76$ +/- 0.12 across n = 2250 images from 15 recordings), but was similar for repeated presentations of the same image (standard deviation across repeats = 0.014). Reconstruction performance was also similar for presentations of the same image in different recordings (standard deviation across recordings = 0.039), despite differences in the population responses and the properties of the RF mosaics (*Figure 2*). The reconstructed images themselves were also very similar

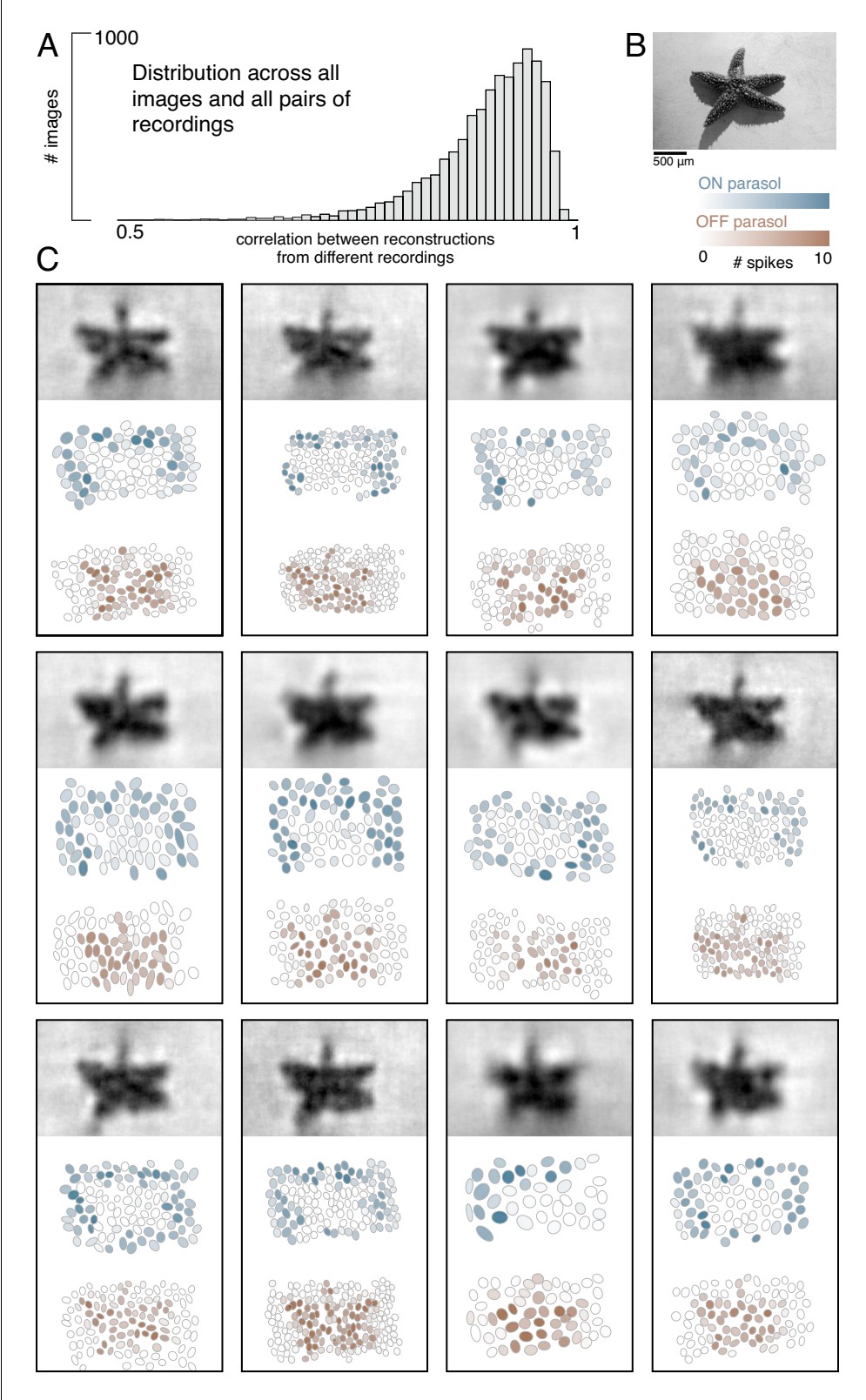

**Figure 2.** Visual representation across retinas. (A) Distribution of correlation between reconstructed images from different recordings, across 150 images and 66 pairs of recordings. (B) Example image. (C) Across 12 recordings, reconstructed images (top, averaged across trials), ON (middle, blue) and OFF (bottom, orange) parasol responses, shown as the mosaic of Gaussian RF fits shaded by the spike count in response to this image. Source files for A are available in *Figure 2—source data 1*.

*Figure 2 continued on next page*

*Figure 2 continued*

The online version of this article includes the following source data for figure 2:

**Source data 1.** Visual representation across retinas.

across recordings (ρ=0.90 +/- 0.06, across 150 images and 66 pairs of recordings; *Figure 2*). The minor differences in performance between recordings were correlated with the average RF size in each recording (ρ=−0.7), which in turn is inversely related to RGC density (*Devries and Baylor, 1997*; *Gauthier et al., 2009*). Qualitatively, large scale image structure seemed to be well captured, but fine details were not. These results indicate that the image structure and the spatial resolution of the RGC population, rather than response variability, were primarily responsible for variation in reconstruction performance across images and recordings.

To further probe the role of the spatial resolution of the RGC population, the reconstructed images were compared to smoothed images, created by convolving the original images with a Gaussian matching the average parasol cell RF size for each recording (see *Figure 1E*, bottom row). Broadly, the smoothed images provided a good approximation to the images obtained by reconstruction. On average, the reconstructed image (averaged across trials) was more similar to the smoothed image than to the original image (ρ=0.91 +/- 0.06 vs. ρ=0.78 +/- 0.11 across n = 2250 images from 15 recordings; p<0.001). The residuals from reconstruction and smoothing, obtained by subtracting the original image, were also similar (ρ=0.83 +/- 0.09), suggesting that reconstruction and smoothing captured and discarded similar features of the original images. While smoothed images do not represent a strict upper limit on reconstruction performance, this analysis further indicates that the RGC density is an important factor in image reconstruction.

Spike latency was also tested as a measure of population response. Spike latency has been shown to convey more stimulus information than spike counts in salamander RGCs in certain conditions (*Gollisch and Meister, 2008*). The RGC response was defined as the time from the image onset to the time of the first spike. This latency response measure led to less accurate reconstruction performance overall (reconstruction from ON and OFF parasol cell responses: Δρ=−0.10 +/- 0.12 across 4500 images from 15 recordings, p<0.001; reconstruction from ON and OFF midget cell responses: −0.16 +/- 0.19 across 3300 images from 11 recordings, p<0.001), although it did improve performance for reconstruction from ON parasol cells alone in two recordings (Δρ=0.04 +/- 0.12 across 600 images from two recordings, p<0.001) and from ON midget cells alone in one recording (Δρ=0.02 +/- 0.1 across 300 images, p<0.001).

## The visual message conveyed by RGCs

To understand how the visual message conveyed by a single RGC depends on the signals transmitted by others, reconstruction was performed from a given cell alone or with other cells of the same type. Cells of the same type exhibited similar response properties (*Chichilnisky and Kalmar, 2002*), with non-overlapping RFs forming a mosaic tiling visual space (*Figure 2*). When a single cell was used for reconstruction, its reconstruction filter (*Figure 3A*, top) was much wider than its spatial RF (*Figure 3A*, bottom, measured with white noise; see Materials and methods), or the spatially localized filter obtained in the full population reconstruction described above (*Figure 1C*). The full width at half maximum of the average single-cell reconstruction filter was roughly four times the average RF width (3.6 +/- 1.4 across 15 recordings). As additional RGCs of the same type were included in reconstruction, the spatial spread of the primary cell's reconstruction filter was progressively reduced, leveling off to a value slightly higher than the average RF size when the six nearest neighbors were included (1.3 +/- 0.2 across 15 recordings; average filters shown in *Figure 3C*, widths shown in *Figure 3D*).

Both the spatial spread of the single-cell reconstruction filter and its reduction in the context of the neural population can be understood by examining how the optimal filters (*Equation 1*) depend on the statistics of the stimulus (S) and response (R). The matrix $R^T R$ represents correlations in the activity of different RGCs. The matrix $R^T S$ represents unnormalized, spike-triggered average (STA) images, one for each RGC. These natural image STAs were broad (*Figure 3A*, top), reflecting the strong spatial correlations present in natural scenes (*Figure 3B*).

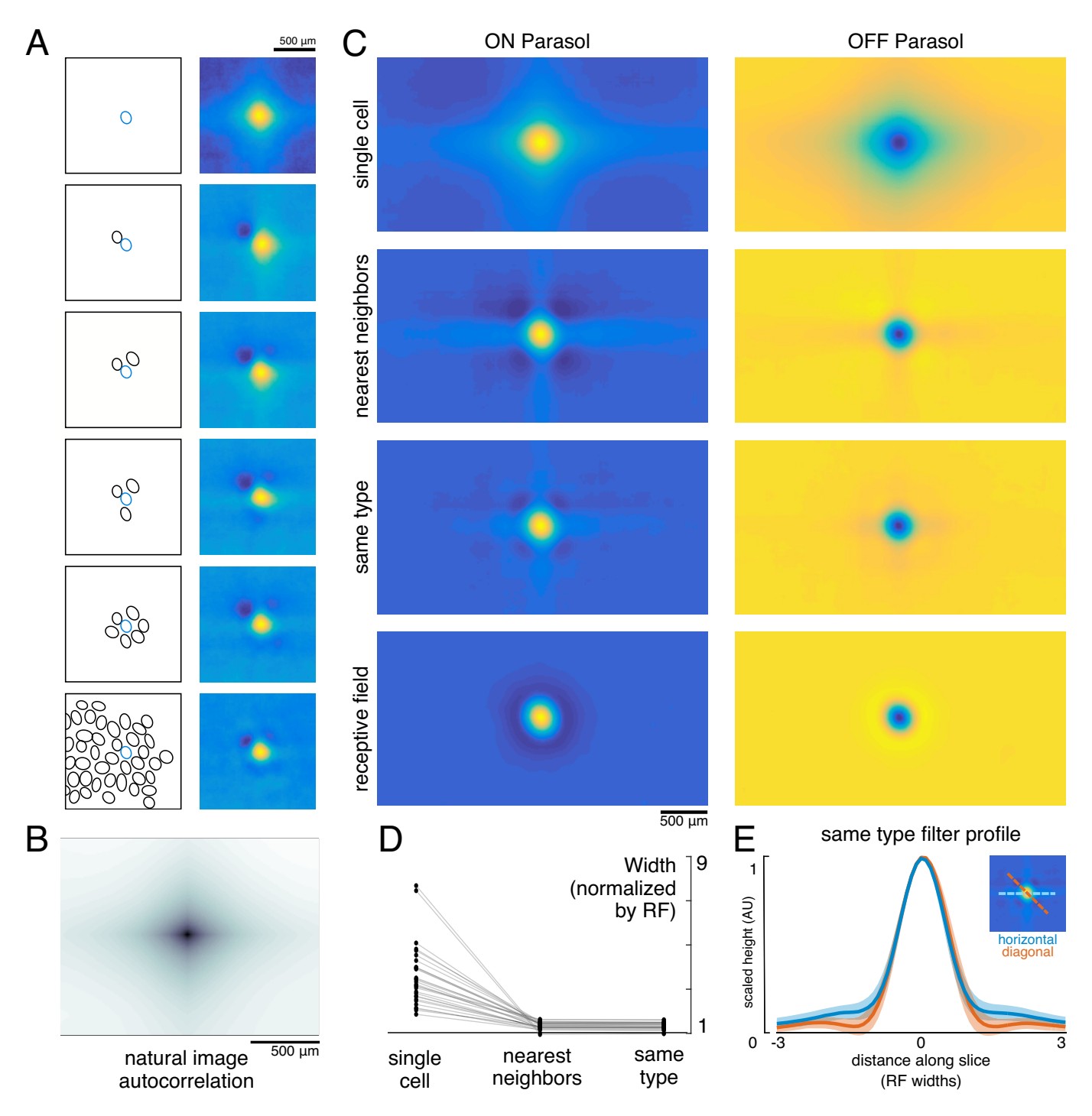

**Figure 3.** Effect of the population on the visual message. (**A**) The reconstruction filter of a single cell as more neighboring cells are included in the reconstruction. Left: receptive fields (RFs) of cells in reconstruction, with the primary cell indicated in blue. Right: Filter of the primary cell. (**B**) Autocorrelation structure of the natural images used here. (**C**) Average ON (left) and OFF (right) parasol cell filters for a single recording. From top to bottom: reconstruction from a single cell, reconstruction from that cell plus all nearest neighbors, reconstruction from that cell plus all cells of the same type, and that cell's RF. (**D**) Filter width, normalized by the RF width. (**E**) Profiles of the same type filters in the horizontal (orange) and diagonal (blue) directions. Average (bold) +/- standard deviation (shaded region) across recordings. Source files for D and E are available in *Figure 3—source data 1*. The online version of this article includes the following source data for figure 3:

**Source data 1.** Effect of the population on the visual message.

For reconstruction from a single cell's responses, $R^TR$ is a scalar, and therefore the single-cell reconstruction filter is directly proportional to the natural image STA. However, in the case of reconstruction from the population, $R^TR$ is a matrix that shapes the reconstruction filter based on the activity of other cells. Specifically, each cell's filter is a linear mixture of its own natural image STA and those of the other cells in the population reconstruction, weighted negatively based on the magnitude of their correlated activity. This mixing resulted in the reduction in the width of the reconstruction filter of a given RGC when nearby cells of the same type were included (*Figure 3C*).

When the complete population of RGCs of the same type was included in the reconstruction, the resulting spatially localized filters were similar to the RFs obtained with white noise stimuli (ρ=0.78 +/- 0.10, n = 997 ON and 1228 OFF parasol cells from 15 recordings). However, some natural image spatial structure remained and was consistent across recordings, cells, and cell types. Most strikingly, the reconstruction filters exhibited broad vertical and horizontal structure (*Figure 3C,E*). This is a known feature of natural scenes (*Girshick et al., 2011*), and is present in the images used here (*Figure 3B*).

In addition, the visual scene was more uniformly covered by the reconstruction filters than by the RFs (*Figure 4A,C*). Coverage was defined as the proportion of pixels that were within the extent of exactly one cell's filter. The filter extent was defined by a threshold, set separately for the reconstruction filters and for the RFs to maximize the resulting coverage value. Across both the ON and OFF parasol cells in 12 recordings, the average coverage was 0.62 +/- 0.06 for the RFs and 0.78 +/- 0.03 for the reconstruction filters (*Figure 4C*; p<0.001). By comparison, expanded RFs, scaled around each RGC's center location to match the average filter width, led to a small reduction in coverage (0.57 +/- 0.06; p<0.001) due to increased overlap. This indicates that the filters are not simply broader versions of the RF, but rather that they are distorted relative to the RFs to fill gaps in the mosaic.

To understand how the differences between reconstruction filters and RFs affected the reconstructed images, reconstruction was performed using the spatial RFs in place of the filters (each RF independently scaled to minimize MSE, see Materials and methods; *Figure 4B*). This manipulation reduced reconstruction performance by 24% (Δρ=−0.12 +/- 0.09 across 4500 images from ON and OFF parasol cells in 15 recordings; p<0.001; *Figure 4D*), primarily in the lower spatial frequencies, which also contain most of the power in the original images (*Figure 4E*). The resulting images were noticeably less smooth in appearance than the optimally reconstructed images, and exhibited structure resembling the RGC mosaic (*Figure 4B*). Thus, although the reconstruction filters generally resembled the RFs, the additional spatial structure related to natural images and the spatial arrangement of RGCs led to smoother reconstructed images. These features may help explain the high consistency in reconstruction performance across many retinas (see above; *Figure 2*).

## Distinct contributions of major cell types

The visual message transmitted by RGCs of a particular type could additionally be affected by the other cell types encoding the same region of visual space (*Warland et al., 1997*). To test this possibility, reconstructions were performed using the responses of a single RGC alone (the primary cell), or in combination with each of the four major cell type populations. For each combination, the reconstruction filters of the primary cells were averaged across all cells of the same type for each recording (*Figure 5A*). Inclusion of all cells of any one cell type reduced the magnitude of the primary cell's reconstruction filter (*Figure 5B*, left). This can be understood by noting that the entries in $(R^TR)^{-1}$, which mix the natural image STAs to produce the reconstruction filters, have the opposite sign of the response correlations. As expected, the correlations were positive for same-polarity cells and negative for opposite-polarity cells (not shown; *Greschner et al., 2011*; *Mastronarde, 1983*). Therefore, the cell's reconstruction filter was reduced in magnitude by positively weighted cells of the opposite polarity, and by negatively weighted cells of the same polarity.

As discussed previously, for parasol cells, inclusion of the remaining cells of the same type substantially reduced the spatial extent of the primary cell's filter (*Figure 3*). However, this did not occur when cells of other types were included in reconstruction instead (*Figure 5B*, right, top two rows). Specifically, the inclusion of the midget cells with the same polarity only slightly reduced the spatial extent of the parasol cell's filter, and inclusion of opposite polarity cells of either type had little effect. This is likely because the other cell types provide roughly uniform coverage, whereas the remaining cells of the same type have a gap in the location of the primary cell, resulting in significant

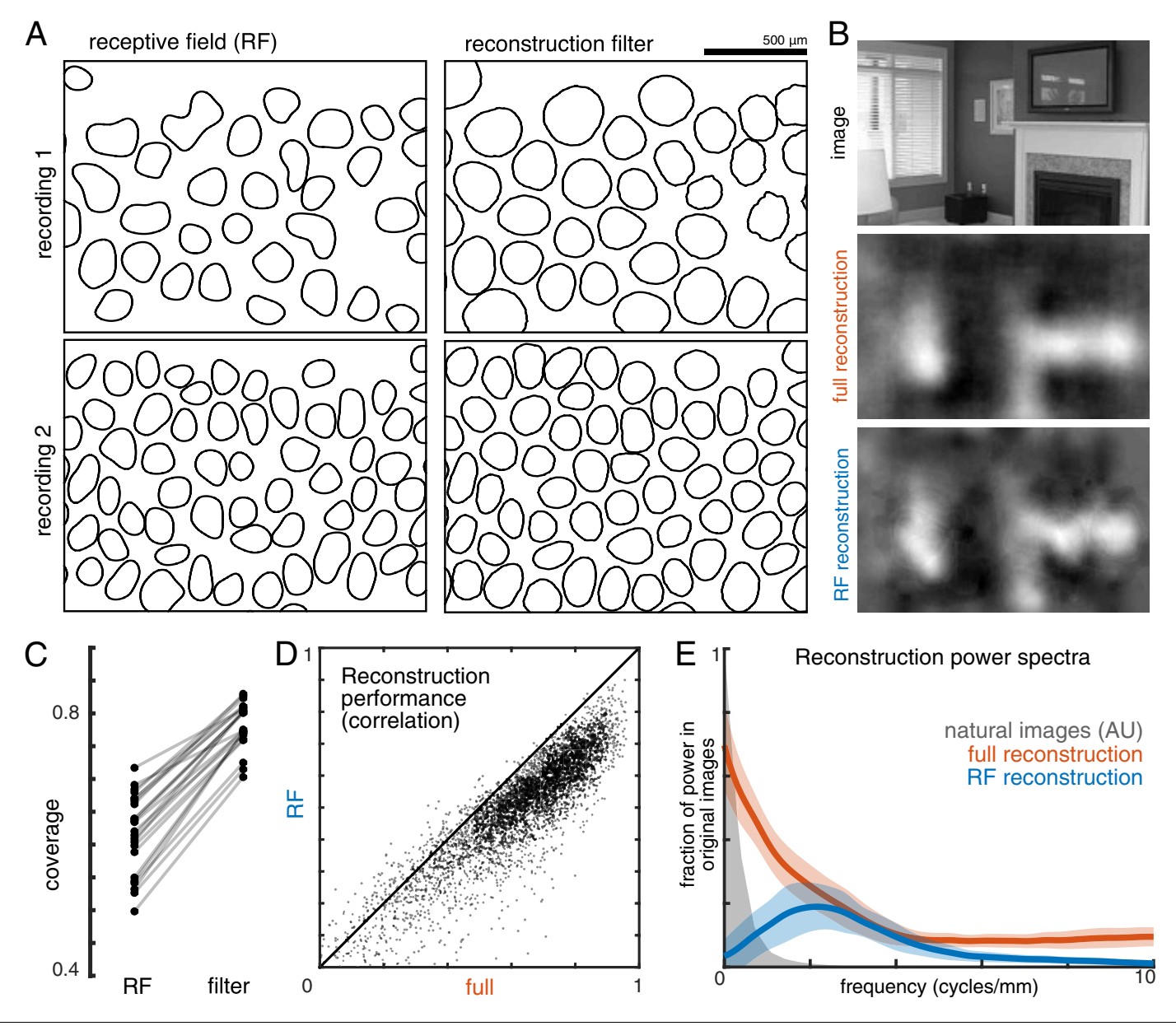

**Figure 4.** Effect of visual message on reconstruction. (**A**) Receptive field (RF, left) and reconstruction filter (right) contours for two sample recordings. (**B**) Reconstruction of an image (top) using the full, fitted filters (middle) and using scaled RFs (bottom). (**C**) Comparison of RF and filter coverage for ON and OFF parasol cells across 12 recordings. (**D**) Comparison of reconstruction performance using scaled RFs or using full, fitted filters, across n = 4800 images from eight recordings. (**E**) Power in the reconstructed images (as a fraction of power in the original image) using fitted filters (orange) or scaled RFs (blue). Average (bold) +/- standard deviation (shaded region) across eight recordings. The original power structure of the natural images is shown in gray and has arbitrary units. Source files for C, D, and E are available in *Figure 4—source data 1*.

The online version of this article includes the following source data for figure 4:

**Source data 1.** Effect of the visual message on reconstruction.

shaping by the immediately neighboring cells. In summary, the spatial structure of the visual message of a single parasol cell is primarily influenced by neighboring cells of the same type and is largely unaffected by cells of other types.

The filters for the midget cells were also shaped by the inclusion of the remaining cells of the same type (*Figure 5A*, second column) and were largely unaffected by the inclusion of opposite polarity cells of either type. However, unlike parasol cells, midget cell filters were significantly

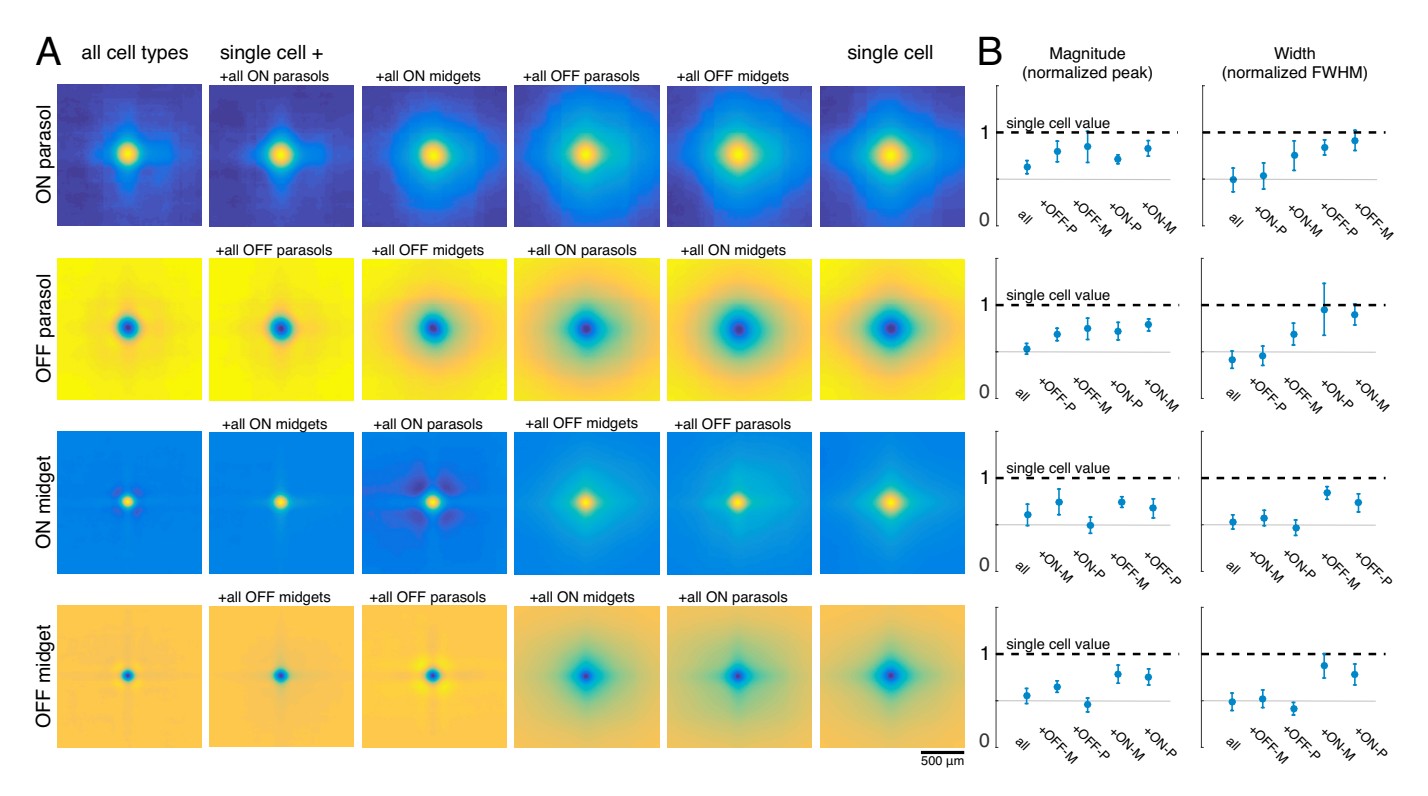

**Figure 5.** Effect of other cell types on the visual message. (**A**) Average reconstruction filters for ON parasol (top row), OFF parasol (second row), ON midget (third row), and OFF midget (bottom row) cells for one recording. Left to right: including all cell types, all cells of the same type, all cells of the same polarity but opposite class, all cells of the opposite polarity but the same class, all cells of opposite polarity and class, and no other cell types. (**B**) Comparison of magnitude (left) and width (right) of average reconstruction filters across conditions, normalized by the features of the single-cell filter. Average +/- standard deviation across recordings is plotted (parasol: n = 11 recordings, midget: n = 5 recordings). Rows correspond to cell types as in A. Source files for B are available in *Figure 5—source data 1*.

The online version of this article includes the following source data for figure 5:

**Source data 1.** Effect of other cell types on the visual message.

affected by the inclusion of the same-polarity parasol cells (*Figure 5A*, third column). This is consistent with known correlations between these cell types (*Greschner et al., 2011*), and the asymmetry may be due to the fact that parasol cells tended to have much stronger responses to the natural images than midget cells. Thus, the interpretation of the visual signal from a midget cell does depend somewhat on the signals sent by the same-polarity parasol cell population.

The image features represented by each cell type were revealed by analysis of the reconstructed images. In particular, the separate contributions of ON and OFF cells, and of parasol and midget cells, were investigated.

To estimate the contribution of ON and OFF cells, reconstruction was performed with ON or OFF parasol cells alone and in combination (*Figure 6A,B*). Reconstructions using just OFF parasol cells were slightly more accurate than using just ON cells, but both were less accurate than reconstruction using the two types together (*Figure 6C*, both: ρ=0.76 +/- 0.12, ON: ρ=0.64 +/- 0.16, OFF: ρ=0.67 +/- 0.14, across n = 2250 images from 15 recordings; all p<0.001). Reconstruction using just ON cells failed to accurately capture intensity variations in dark areas of the image, while reconstruction with just OFF cells failed to capture variations in light areas of the image (for pixel values above the mean value: ρ=0.57 for ON and 0.26 for OFF, for pixel values below the mean value: ρ=0.31 for ON and 0.68 for OFF). Only a narrow middle range of pixel intensities were effectively encoded by both types (*Figure 6D*). This is consistent with known output nonlinearities, which suppress responses to stimuli of the non-preferred contrast, and therefore limit linear reconstruction in that range. Thus, both ON and OFF cells were necessary to reconstruct the full range of image contrasts.

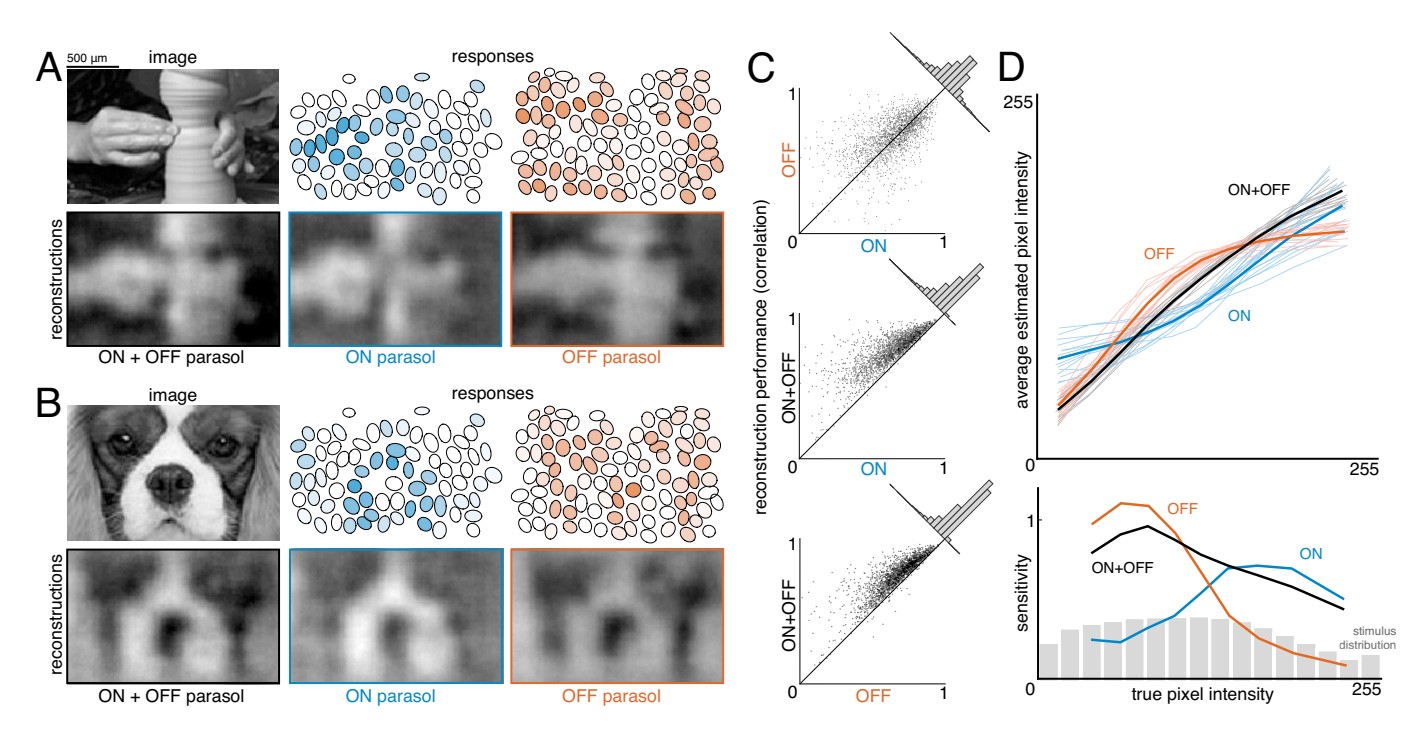

**Figure 6.** Contributions of ON and OFF parasol cells. (A,B) Example images, responses, and reconstructions from ON and OFF parasol cells. Top left: original image. Top right: Parasol cell mosaics shaded by their response value (ON - blue, middle, OFF - orange, right). Bottom left: reconstruction from both cell types. Bottom right: reconstruction from just ON (blue, middle) or just OFF (orange, right) parasol cells. (C) Reconstruction performance for ON vs. OFF (top), both vs. ON (middle), and both vs. OFF (bottom), with n = 2250 images from 15 recordings. (D) Average estimated pixel intensity (top) and sensitivity (bottom, defined as Δaverage estimated pixel intensity/Δtrue pixel intensity) vs. true pixel intensity for ON (blue), OFF (orange), and both (black). Individual recordings are shown in the top plot, with the average in bold. Source files for C and D are available in *Figure 6—source data 1*.

The online version of this article includes the following source data for figure 6:

**Source data 1.** Contributions of ON and OFF parasol cells.

Reconstruction using the responses of both cell types seemed to encode darker pixels more accurately than lighter pixels (*Figure 6D*, bottom panel, black curve), consistent with the reconstruction performance from each type separately. This could reflect the fact that ON cells are less dense (*Chichilnisky and Kalmar, 2002*), and/or the fact that the natural image distribution is skewed towards darker pixel values (*Figure 6D*, bottom panel, gray distribution), potentially placing greater weight on the accurate reconstruction of these values. In addition, ON cells exhibit a more linear contrast-response relationship (*Chichilnisky and Kalmar, 2002*), so there is less reconstruction performance difference between preferred and non-preferred contrasts.

To estimate the contributions of parasol and midget cells, reconstruction was performed using parasol cells or midget cells or both (*Figure 7A,B*). As expected, reconstruction using both parasol and midget cells was more accurate than using either alone (*Figure 7C*, both: ρ=0.81 +/- 0.10, parasol: ρ=0.77 +/- 0.12, midget: ρ=0.73 +/- 0.13, across n = 1050 images from seven recordings; all p<0.001). Images reconstructed from midget cells contained more high-frequency spatial structure, consistent with their higher density (*Figure 7D*). However, the images reconstructed from parasol cells had 50% higher signal-to-noise (defined as standard deviation across images/standard deviation across repeats), resulting in the slightly higher reconstruction performance from parasol cells.

The above analysis obscures the significantly different temporal responses properties of these two cell classes. In particular, parasol cells have more transient responses (*de Monasterio, 1978*; *De Monasterio and Gouras, 1975*; *Gouras, 1968*), which may allow them to convey information more rapidly than midget cells. To test this possibility, image reconstruction was performed using

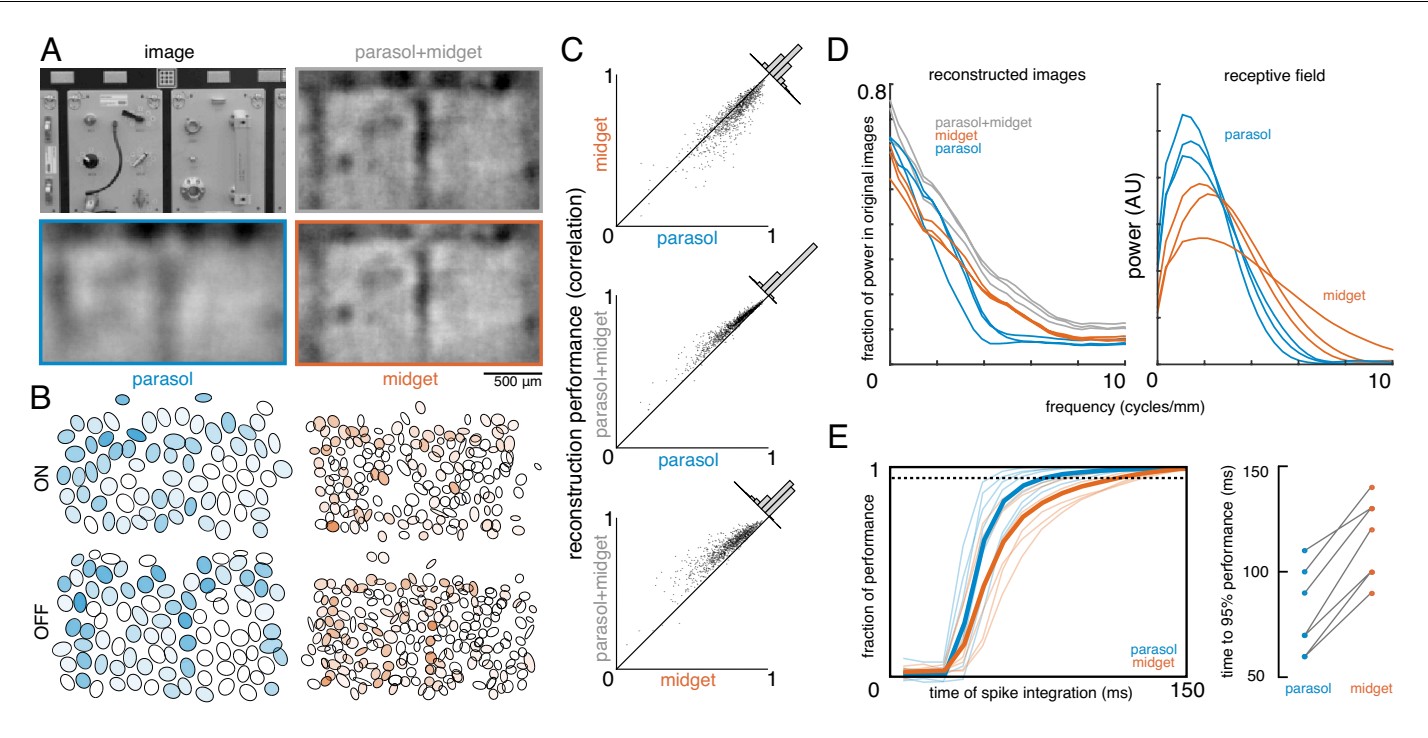

**Figure 7.** Contributions of the parasol and midget cell classes. (A) Example image and reconstructions for parasol and midget cells. Top left: original image. Top right: reconstruction with parasol and midget cells (gray). Bottom left: reconstruction with only parasol cells (blue). Bottom right: reconstruction with only midget cells (orange). (B) Cell type mosaics shaded by their response values, for ON (top) and OFF (bottom) parasol cells (left, blue) and midget cells (right, orange). (C) Reconstruction performance for midget vs. parasol (top), both vs. parasol (middle), and both vs. midget (bottom). (D) Power in the reconstructed images as a fraction of power in the original image (left) and receptive fields (right) for parasol cells (blue), midget cells (orange), and both types (gray) for each of three recordings. (E) Left: Fraction of peak reconstruction performance with increasing spike integration times for parasol (blue) and midget (orange) cells, with averages across recordings shown in bold. Dotted line indicates 95% performance. Right: Time to 95% performance for parasol and midget reconstructions across seven recordings. Source files for C, D, and E are available in *Figure 7—source data 1*.

The online version of this article includes the following source data for figure 7:

**Source data 1.** Contributions of the parasol and midget cell classes.

spikes collected over increasing windows of time after the image onset. The reconstruction performance of parasol cells increased quickly and reached 95% of peak reconstruction performance at 80 +/- 20 ms, while the performance of midget cells increased more slowly, and reached 95% performance at 116 +/- 19 ms (across seven recordings; *Figure 7E*). This difference indicates that spatio-temporal reconstruction will be necessary to fully reveal the distinct contributions of these two classes (see Discussion).

## The effect of correlated firing

The above results indicate that the visual message of each RGC, and the contributions of each cell type, are shaped by correlated activity. However, these analyses do not distinguish between stimulus-induced (signal) correlations, and stimulus-independent (noise) correlations that arise from neural circuitry within and across cell types in the primate retina (*Greschner et al., 2011*; *Mastronarde, 1983*).

To test the effect of noise correlations, reconstruction performance was evaluated on repeated presentations of test images. This performance was compared to a control condition in which the responses of each cell were independently shuffled across trials to remove noise correlations while preserving single-cell statistics and signal correlations. The reconstruction filters (computed from unshuffled training data) were then used to reconstruct the test images, using either the shuffled or

unshuffled responses. In principle, shuffling could result in a net increase or decrease in reconstruction accuracy, due to two opposing factors. Because the reconstruction filters incorporate the correlated activity present in training data (*Equation 1*), any deviation from this correlation structure in the test data could reduce performance. On the other hand, if noise correlations produce spatial structure in the reconstructions that obscures the structure of the natural images, their removal could enhance reconstruction performance. The relative influence of these competing effects could also depend on the overall fidelity of the reconstruction.

Accordingly, the shuffling manipulation was tested using three response measures. In the first, RGC responses were calculated by counting spikes in the 150 ms window after image onset, as above. In the second, the response was measured at the intrinsic time scale of correlations (~10 ms; *DeVries, 1999*; *Mastronarde, 1983*; *Meister et al., 1995*; *Shlens et al., 2006*), by counting spikes in fifteen 10 ms bins, and reconstructing with this multivariate response vector instead of the scalar spike count. In the third, spikes were counted only in the 10 ms bin that had the highest average firing rate (50–60 ms after image onset). While the third approach did not utilize all the available information in the responses, it was used to mimic low-fidelity or rapid perception scenarios, which would have fewer stimulus-driven spikes available for reconstruction.

Reconstruction using the first two response measures had similar unshuffled performance ($\rho=0.76$ +/- 0.12 and 0.75 +/- 0.12 respectively), and low variation across trials (standard deviation across repeats = 0.015). With these measures, shuffling had a very small and detrimental effect on reconstruction (across 3 recordings with 27 repeats of 150 test images: (1) $\Delta\rho=-0.0004$ +/- 0.0017; $|\Delta\rho|$ =0.0012 +/- 0.0012; $p<0.001$, (2) $\Delta\rho=-0.0008$ +/- 0.0019; $|\Delta\rho|=0.0014$ +/- 0.0015; $p<0.001$). In each case, the magnitude of the change in correlation represented about 10% of the variation in reconstruction accuracy across trials, which represents roughly how much improvement could be expected (*Figure 8*). For comparison, shuffling the responses in each time bin independently across trials (rather than the responses of each cell independently) had a much larger effect ($\Delta\rho=-0.02$ +/- 0.01), consistent with previous results (*Botella-Soler et al., 2018*), indicating that the autocorrelation structure across time is more important for reconstruction than the noise correlation structure across cells. Thus, in these conditions, noise correlations had a limited impact on reconstruction, regardless of the time scale of analysis.

Reconstruction using the third measure had lower unshuffled performance ($\rho=0.64$ +/- 0.14), and higher variation across trials (standard deviation across repeats = 0.039). In this case, shuffling led to a more consistent, but still small, increase in reconstruction performance ($\Delta\rho=0.0071$ +/- 0.0076; $|\Delta\rho|$ =0.0075 +/- 0.0072; $p<0.001$). The increase represented a larger fraction of the variation in reconstruction accuracy across trials (20%; *Figure 8*). This suggests that in low-fidelity, high-noise situations, noise correlations in the RGC population can partially obscure the structure of natural images, even if reconstruction is designed to take the correlations into account.

## Nonlinear reconstruction

Linear reconstruction provides an easily interpretable estimate of the visual message, but it may limit the quality of reconstruction by not extracting all the information available in the neural responses and may also differ greatly from how the brain processes the retinal input. Therefore, two simple extensions of linear reconstruction were tested: transformation of the responses using a scalar nonlinearity, and inclusion of interaction terms between nearby cells.

In the first case, the response of each cell was transformed using a scalar nonlinearity, and linear regression (*Equation 1*) was performed to reconstruct images from the transformed response. The stimulus estimate $S_{NL}$ is given by $S_{NL} = f(R) \cdot W_{NL}$, where $W_{NL}$ is a matrix of reconstruction weights (refitted using the transformed responses), and $f(R)$ is the scalar nonlinear transform of the population response vector $R$. This is equivalent to inverting a linear-nonlinear (LN) encoding model of the form $R = g(K \cdot S)$, where $g$ is the inverse of $f$, and $K$ is a different set of weights (note that in general a nonlinear encoder may not require an equivalent nonlinear decoder for optimum performance; see *Rieke et al., 1997* for a full discussion). A common form of the LN encoding model uses an exponential nonlinearity, $g = \exp()$; therefore, the inverse function $f = \log()$ was used for reconstruction, and the response for each cell was defined as the spike count plus 1. A square root transformation was also tested, and yielded similar results (not shown).

The relationship to pixel values was more linear for the transformed RGC responses than for the original responses ($\Delta$linear fit RMSE = $-1.9$ +/- 1.5 across n = 2225 cells from 15 recordings;

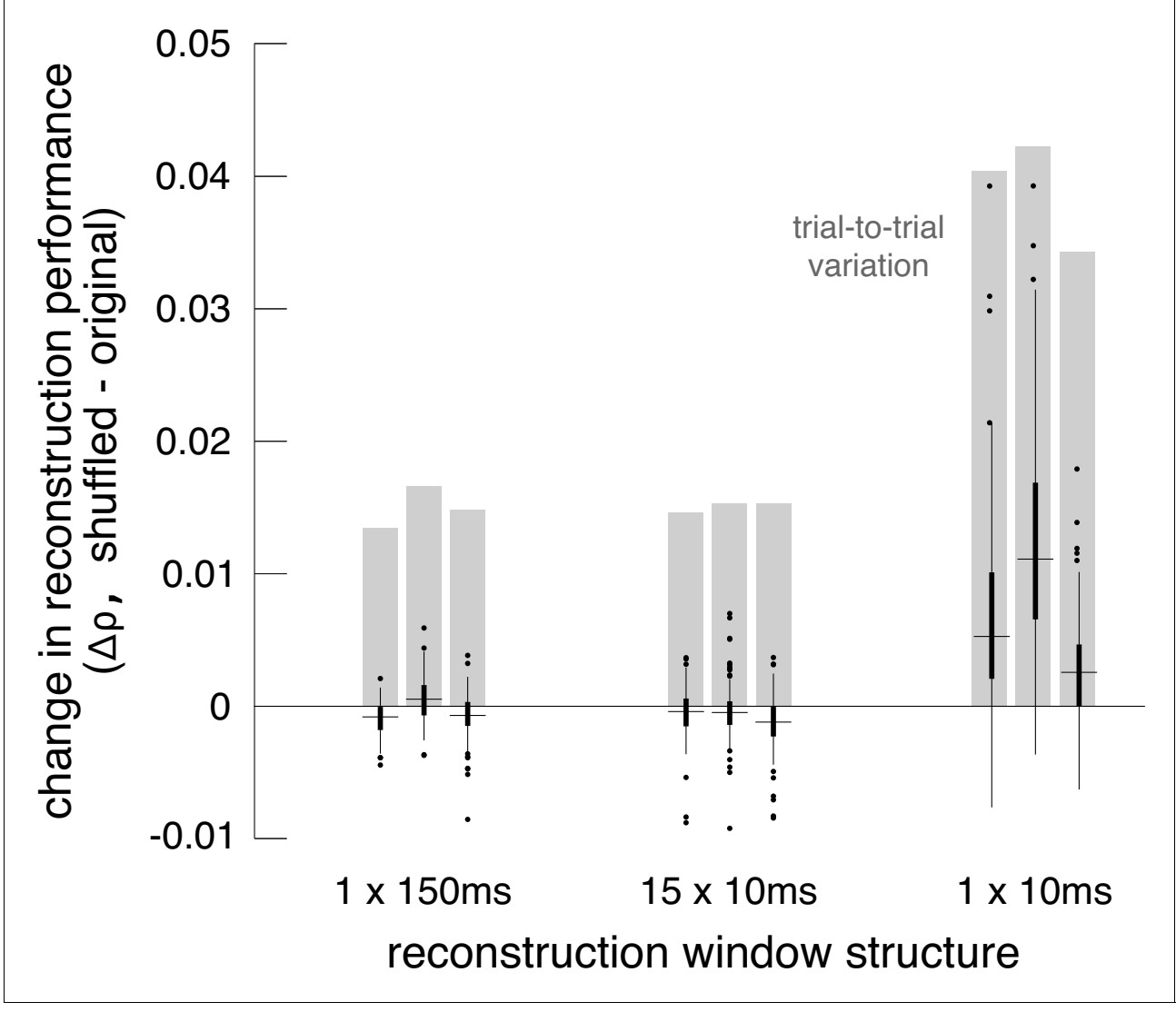

**Figure 8.** Effect of noise correlations. The change in reconstruction performance (Δρ) when using shuffled data for three scenarios: one 150 ms window, fifteen 10 ms windows, and one 10 ms window. Black bars show median +/- interquartile range for three recordings (each shown separately). Gray bars show the standard deviation in the reconstruction performance across trials. Source files are available in *Figure 8—source data 1*.

The online version of this article includes the following source data for figure 8:

**Source data 1.** Effect of noise correlations.

*Figure 9A,B*), indicating that this inverse function captured at least some of the nonlinearity in retinal signals. The nonlinear transformation slightly increased reconstruction accuracy when using the responses of ON or OFF parasol cells alone (across 15 recordings with 300 images each: ON parasol: Δρ=0.013 +/- 0.051, p<0.001; OFF parasol: Δρ=0.015 +/- 0.035, p<0.001; *Figure 9C*). However, it did not help when using the responses of ON and OFF parasol cells together (Δρ=−0.0017 +/- 0.032, p=0.001; *Figure 9C*). This likely reflects the fact that the relationship between the true pixel values and the pixel values reconstructed using the original, untransformed responses was already approximately linear when using both cell types, but not when using just one cell type (*Figure 6*). In addition, using the raw responses of both cell types was more effective than using the transformed responses of either type alone (ON parasol: Δρ=−0.09 +/- 0.1, p<0.001; OFF parasol: Δρ=−0.06 +/- 0.1, p<0.001), suggesting that intensity information cannot be directly recovered fully from either ON or OFF cells alone.

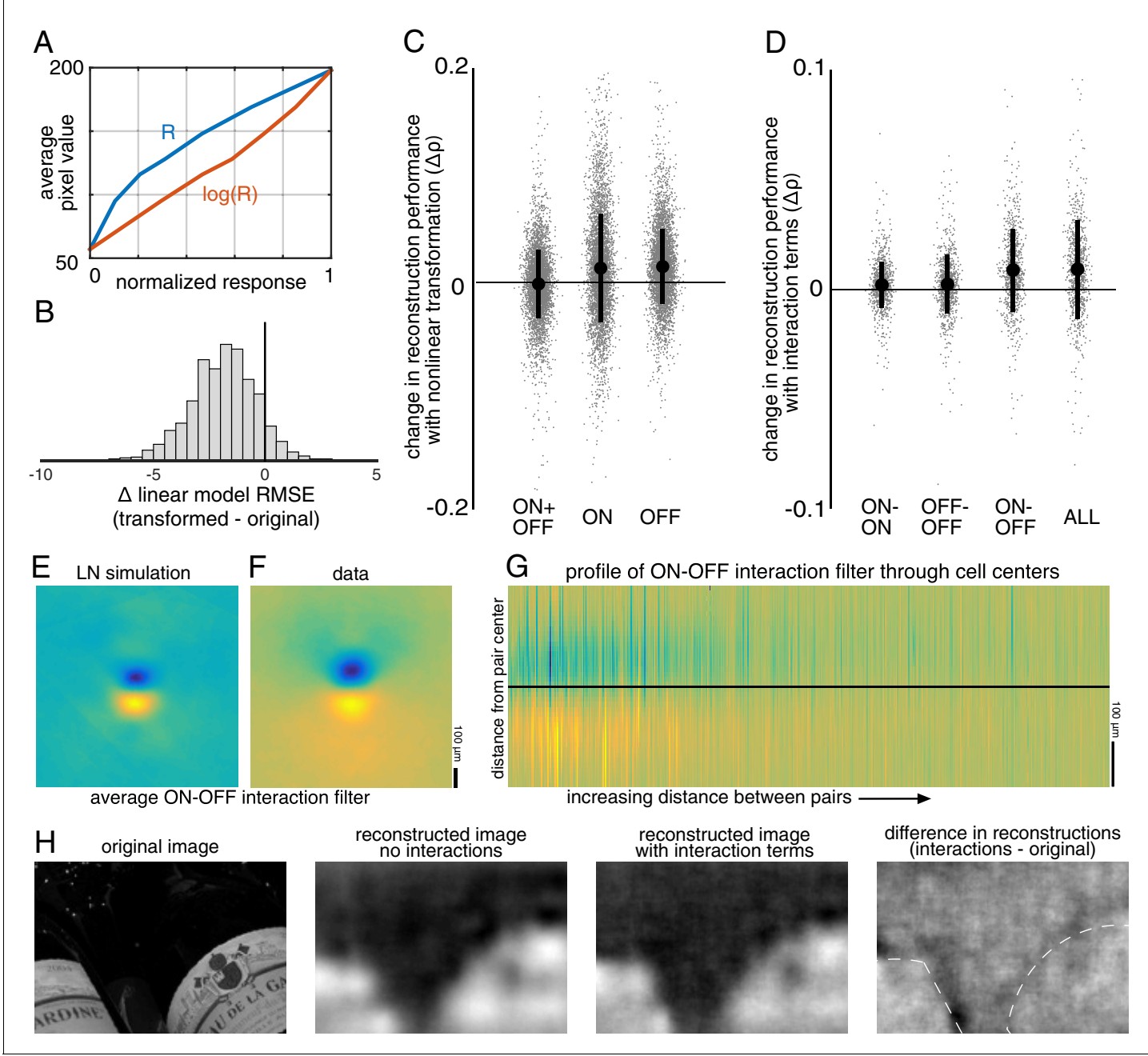

**Figure 9.** Nonlinear reconstruction. (**A**) Average pixel value in receptive field center vs. original response (blue) and transformed response (orange). (**B**) Distribution (across n = 2225 cells from 15 recordings) of the change in RMSE of a linear model (mapping from response to pixel value) when using the transformed response. (**C**) Change in reconstruction performance (correlation) when using transformed responses (log(R)) for reconstruction with either ON and OFF parasol cells, only ON parasol cells, or only OFF parasol cells. Individual images (n = 300 from each of the 15 recordings) are plotted in gray with jitter in the x-direction. The black bars represent mean +/- standard deviation, and the standard error is smaller than the central dot. (**D**) Change in reconstruction performance (correlation) when including interaction terms. Individual images (n = 300 from each of the three recordings) are plotted in gray with jitter in the x-direction. The black bars represent mean +/- standard deviation, and the standard error is smaller than the central dot. (**E,F**) Average reconstruction filters corresponding to ON-OFF type interactions, centered and aligned along the cell-to-cell axis, for simulation (**E**) and data (**F**). (**G**) 1D Profiles of all ON-OFF interaction filters through the cell-to-cell axis, sorted by distance between the pair. (**H**) Example image (left), reconstructions with and without interaction terms (middle), and difference between the reconstructions, with dotted lines indicating edges (right). Source files for B, C, and D are available in *Figure 9—source data 1*.

The online version of this article includes the following source data for figure 9:

**Source data 1.** Nonlinear reconstruction.

Nonlinear interactions between the signals from different cells could also potentially increase reconstruction performance. To test this idea, the products of spike counts in pairs of neighboring cells were added as predictors in the linear reconstruction. Neighbors were defined as cells with RF centers that were within 1.5 times the median nearest neighbor distance between RF centers of the cells of the same type. For parasol cells, this definition resulted in roughly 6 ON and 6 OFF neighbors per cell, as expected (see *Figure 2*). Including these interactions produced a small increase in reconstruction accuracy ($\Delta\rho=0.0093$ +/- 0.023, across three recordings with 300 test images each; p<0.001; regularization did not lead to improved performance). The primary contribution was from ON-OFF pairs (ON-OFF: $\Delta\rho = 0.0089$ +/- 0.019, not significantly different than all pairs, p=0.2; ON-ON: $\Delta\rho=0.0021$ +/- 0.010 and OFF-OFF: $\Delta\rho=0.0024$ +/- 0.013, both significantly different than all pairs, p<0.001; *Figure 9D*). The reconstruction filters associated with these interaction terms typically had an oriented structure orthogonal to the line between the RF centers of the two cells (*Figure 9F,G*), suggesting that the improvement in reconstruction may come primarily from using the joint activation of partially overlapping ON and OFF cells to capture edges in the visual scene.

## Comparison to simple models of RGC light response

The above analyses revealed that noise correlations and interactions between cells and cell types had a limited impact on reconstruction performance, suggesting that more complicated features of retinal encoding may not be important for linear reconstruction. To further explore this idea, simple LN models were used to simulate RGC responses across all 15 recordings, and the primary features of reconstructions from recorded and simulated spike trains were compared. The simulated spike count of each RGC in response to a given image was calculated by filtering the image with the spatial RF, and then passing that value through a fitted sigmoidal nonlinearity to obtain a firing rate (see Materials and methods). The noise in the recorded spike counts was sub-Poisson (not shown; see *Uzzell and Chichilnisky, 2004*); therefore, the simulated firing rate was directly compared to the trial-averaged, recorded firing rate. This model captured RGC responses to static images with reasonable accuracy (correlation between simulated and average recorded spike counts: 0.76 +/- 0.13 across n = 997 ON parasol cells; 0.84 +/- 0.09 across n = 1228 OFF parasol cells; see *Chichilnisky, 2001*). Note that by definition, the model incorporated the measured functional organization of the retina, including retina-specific RF mosaic structure and cell-type specific response properties, both of which are necessary to understand the visual message (see above).

Reconstructions with recorded and simulated spike trains revealed broadly similar properties in the filters and reconstructed images. The filters fitted to the recorded and simulated spike trains were similar ($\rho=0.84$ +/- 0.09 across 2225 parasol cells from 15 recordings), and shared key features, such as horizontal and vertical structure (*Figure 10A,C*). The reconstructed images themselves were also similar (correlation between images reconstructed from simulated and recorded spike counts: 0.93 +/- 0.04 across n = 2250 images from 15 recordings; *Figure 10B,C*), as was the reconstruction performance (simulated: $\rho=0.79$ +/- 0.11; recorded: $\rho=0.78$ +/- 0.11; $\Delta\rho=-0.003$ +/- 0.03; across 2250 images from 15 recordings; *Figure 10C*).

The simulated spike trains also replicated the structure of nonlinear interactions between cells. This was observed by using the simulated responses of ON and OFF cells and the products of the responses of neighboring cells, as above, to reconstruct natural images. The spatial reconstruction filter corresponding to the interaction term between nearby ON and OFF cells was oriented and qualitatively similar to the interaction filters obtained with real data (*Figure 9E,F*). However, this was not the case for responses simulated using a linear model without any response rectification (not shown) – in this case, the filter corresponding to the interaction term had no clear structure.

The model reveals that although the visual messages of RGCs depend on their spatial and cell-type specific organization, as well as the statistics of the stimulus, their essential structure can be understood using simple models of RGC encoding. Furthermore, some degree of nonlinear encoding is necessary to explain the oriented interaction filters observed in the data.

## Spatial information in a naturalistic movie

In natural vision, a continuous stream of retinal responses is used to make inferences about the dynamic external world. Therefore, the reconstruction approach above – using the accumulated spikes over a fraction of a second to reconstruct a flashed image – could fail to capture important

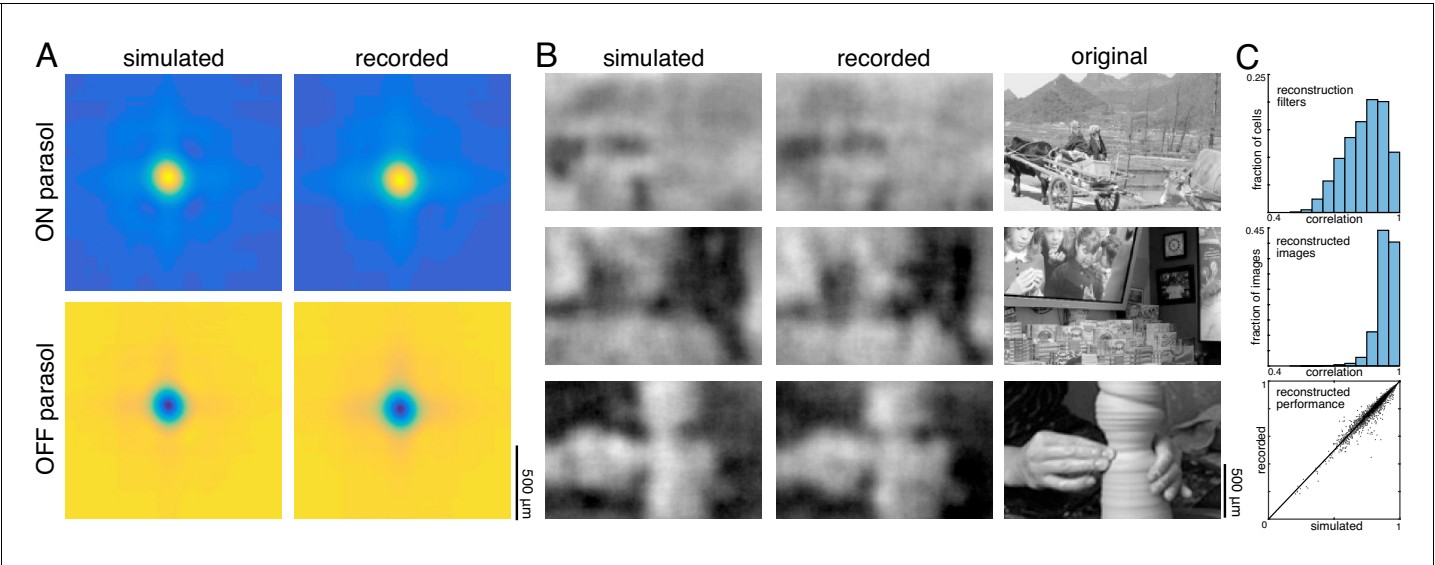

**Figure 10.** Comparison to simulated spikes. (**A**) Average reconstruction filters calculated from spikes simulated using linear-nonlinear models (left) or recorded (right). (**B**) Images reconstructed from simulated (left) or recorded (middle) spikes, compared to the original images (right). (**C**) Comparison of reconstructions with recorded and simulated spike counts: filters (top; $\rho=0.84$ +/- 0.09 across 2225 parasol cells from 15 recordings), reconstructed images (middle; $\rho=0.93$ +/- 0.04 across n = 2250 images from 15 recordings), and performance (bottom; simulated: $\rho=0.79$ +/- 0.11; recorded: $\rho=0.78$ +/- 0.11; $\Delta\rho=-0.003$ +/- 0.03; across 2250 images from 15 recordings). Source files for C are available in *Figure 10—source data 1*. The online version of this article includes the following source data for figure 10:

**Source data 1.** Comparison to simulated spikes.

aspects of normal vision. To test whether the above results extend to spatiotemporal reconstructions, a naturalistic movie, consisting of a continuous stream of natural images with simulated eye movements superimposed, was reconstructed from the spike trains of RGCs. The spike trains were binned at the frame rate of the movie (120 Hz), and linear regression was performed between the frames of the movie and the RGC responses in 15 bins following each frame, resulting in a spatio-temporal reconstruction filter for each RGC.

A spatial summary of the filter for each cell was obtained by first calculating the average time course of the strongest pixels, and then projecting each pixel of the full filter against this time course (examples shown in *Figure 11A*; see Materials and methods). This spatial filter was highly correlated with the spatial reconstruction filters of the same cells obtained in the preceding analysis with flashed images ($\rho=0.87$ +/- 0.07, n = 351 parasol cells from three recordings; *Figure 11B*). The dynamic filters were approximately space-time separable (explained variance from first principal component = 0.85 +/- 0.13). The remaining unexplained variance contained significant apparent structure as well as noise (not shown), which may be important for further understanding spatiotemporal processing in the retina and the underlying mechanisms, but was not explored further (*Benardete and Kaplan, 1997a*; *Benardete and Kaplan, 1997b*; *Dawis et al., 1984*; *Derrington and Lennie, 1982*; *Enroth-Cugell et al., 1983*). The large fraction of variance explained by a space-time separable filter suggests that the essential spatial features of the visual message observed in spatial reconstructions largely extend to spatiotemporal vision. In addition, the reconstructed movie frames were similar to reconstructions of static images (between static reconstruction and average reconstructed frame: $\rho=0.72$ +/- 0.19 across 120 images from three recordings, *Figure 11C*).

## Discussion

Linear reconstruction of natural images was used to investigate the spatial information transmitted to the brain by complete populations of primate RGCs. The quality of the reconstructions was consistent across retinas. The optimal interpretation of the spikes produced by a RGC – that is its visual message – depended not only on its encoding properties, but also on the statistics of natural scenes

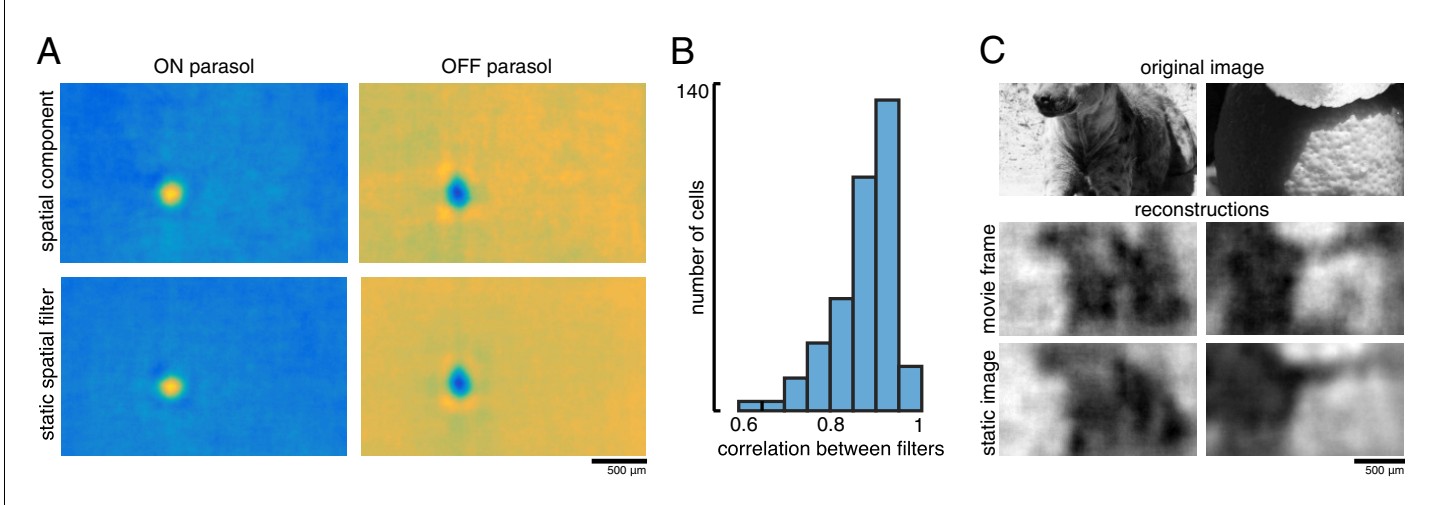

**Figure 11.** Spatiotemporal reconstruction. (**A**) Examples of the spatial components extracted from the spatiotemporal reconstruction filter (top) and the static spatial reconstruction filters (bottom) for an ON (left) and OFF (right) parasol cell. (**B**) Correlation between spatial component and static filter (ρ=0.87 +/- 0.07 across n = 351 cells from three recordings). (**C**) Example reconstructions of movie frames and of static images. Source files for B are available in *Figure 11—source data 1*.

The online version of this article includes the following source data for figure 11:

**Source data 1.** Spatiotemporal reconstruction.

and the spatial arrangement of other RGCs. These factors enabled smoother natural image reconstructions from the RGC population than would be expected from the RFs alone. In addition, the visual representation conveyed by each cell type reflected its distinct encoding properties, and for ON and OFF parasol cells, was largely independent of the contributions of other cell types. Overall, the results were consistent with a simple, linear-nonlinear model of RGC encoding, incorporating the spatial properties, contrast-response properties, and collective functional organization of the four major RGC types. Finally, a limited test of spatiotemporal reconstruction indicated that these results may generalize to natural vision.

The results show that the dependence of a given RGC's visual message on the responses of other RGCs, which was demonstrated previously in the temporal domain using a spatially uniform random flicker stimulus (*Warland et al., 1997*), extends to the spatial domain in natural viewing conditions. For decades, the spatial visual message of a RGC has been estimated using its RF, measured with artificial stimuli. However, due to spatial correlations in natural scenes, the response of a RGC contains information about the stimulus far beyond its RF. In this light, it is at first surprising that the visual message is spatially localized and similar to the classical RF (*Figure 3A,C*). However, nearby regions of visual space are already 'covered' by the neighboring RGCs of the same type, and the redundant information in adjacent cells apparently contributes little to representing the image structure. Even so, the visual messages retain some explicit horizontal and vertical natural scene structure, and collective spatial organization, not present in the RFs. This structure results in smoother reconstructions and more uniform coverage of visual space than the coverage provided by the RF mosaic (*Figure 4*). In this sense, the visual message of each RGC differs from its RF, specifically in a way that reflects its coordination with other nearby cells. The significance of natural scene statistics for interpreting the neural code has also been suggested in the visual cortex (*Naselaris et al., 2009*), and can be used as a prior to improve image estimates in multi-step reconstruction methods (*Parthasarathy et al., 2017*).

Each of the major RGC types conveyed distinct visual representations, consistent with their encoding properties. For the most part, these were independent of the contributions of the other types, indicating that the major primate RGC types, despite covering the same region of visual space, conveyed different stimulus features. However, this separation was clearer for the ON and OFF types than for the parasol and midget cell classes, because the midget cell filters were

influenced by the inclusion of same-polarity parasol cells. Further analysis in the temporal domain (see *Figure 7E*) may be necessary to clarify the separation of these two classes. Both ON and OFF cell types were necessary to reconstruct the full contrast range of the images, because responses from a single cell type resulted in less accurate reconstructions even if they were linearized. It is not clear why the retina separates visual information into separate cell type channels. The roughly linear intensity representation by ON and OFF cell types together (but not individually) is consistent with suggestions that encoding by multiple cell types with nonlinear response properties could enable relatively simple linear reconstruction by downstream neurons (*DiCarlo et al., 2012*; *Gjorgjieva et al., 2019*). There also may be more complicated interactions between different cell types that another reconstruction method could reveal. As new cell types are identified and characterized (*Puller et al., 2015*; *Rhoades et al., 2019*), their contributions to vision may be more fully revealed by these linear and simple nonlinear reconstruction approaches.

Overall, the results presented here were consistent with predictions from a simple, independent pseudo-linear model for RGC light responses, despite known nonlinearities and correlations in the retinal circuitry. Specifically, replacing the recorded spike trains with simulated spike trains, generated by LN models fitted to each RGC, resulted in similar reconstruction filters and reconstructed images (*Figure 10*). Obviously, the LN model by itself cannot explain the many features of encoding observed here; instead, the specific spatial properties, contrast-response properties, and collective organization of the major RGC types captured in the present measurements are crucial for understanding the structure of the visual message. The similarity of reconstruction from LN models and recorded data is consistent with the limited impact of interaction terms and stimulus-independent (noise) correlations, the importance of which has been debated (*Cafaro and Rieke, 2010*; *Ganmor et al., 2015*; *Meytlis et al., 2012*; *Nirenberg et al., 2001*; *Pillow et al., 2008*; *Puchalla et al., 2005*; *Ruda et al., 2020*; *Zylberberg et al., 2016*). While the impact of noise correlations on reconstruction in the present data was limited by the low total noise in the accumulated spike counts, this may not reflect natural vision, in which perception and action occur too quickly to utilize all the stimulus-driven spikes from each RGC, and sometimes must rely on visual inputs with low light levels or spatial contrast (*Ruda et al., 2020*). A low-fidelity situation was mimicked by reducing the spike integration time window to 10 ms, a manipulation that revealed an increased but still small effect of noise correlations. It is also possible that these results would be affected by removing noise correlations from both the training and testing data, but evaluating this possibility would require longer repeated presentations of training stimuli than were performed here.

It is uncertain how close the reconstructions presented here are to the best possible reconstructions given the data, and how much additional information could potentially be extracted from the spike trains. Acuity has been shown to track with midget cell RF size (*Dacey, 1993*; *Merigan and Katz, 1990*; *Rossi and Roorda, 2010*; *Thibos et al., 1987*), indicating that the reconstructions shown in *Figure 7* may accurately represent the quality of visual information transmitted to the brain. In addition, it has been suggested that simple decoders may be sufficient, even when the encoding is highly nonlinear (*DiCarlo et al., 2012*; *Gjorgjieva et al., 2019*; *Naselaris et al., 2011*; *Rieke et al., 1997*). However, alternative approaches may be worth exploring, and could extract additional information. For example, different measures of response, such as latency (*Gollisch and Meister, 2008*; *Gütig et al., 2013*) and relative activity (*Portelli et al., 2016*), have been shown to convey more stimulus information than spike counts for non-primates under some conditions. This was not the case in the present data, which may be due to high-maintained firing rates in the mammalian retina (*Troy and Lee, 1994*; see *Figure 1B*), which make it difficult to identify the first stimulus-driven spike. In addition, recent studies have indicated that nonlinear and deep learning models could improve reconstruction performance for static images, moving patterns, and naturalistic movies (*Botella-Soler et al., 2018*; *Kim et al., 2020*; *Parthasarathy et al., 2017*; *Zhang et al., 2020*). While these approaches make the visual message more difficult to define, they could be used to extract richer information potentially present in RGC responses. Models that are interpretable while allowing for some nonlinearities could also be used to further investigate the visual message (*Pillow et al., 2008*).

Attempting to extract more sophisticated visual information may also reveal additional information conveyed by RGCs, for example, by expanding to more complex, dynamic natural stimuli. Spatiotemporal stimuli, which were only explored here in a limited way, and/or chromatic stimuli, could further illuminate the impact of spike timing, the encoding of dynamic and space-time inseparable

features, and the distinct roles of the multiple cell types (*Benardete and Kaplan, 1997a*; *Benardete and Kaplan, 1997b*; *Berry et al., 1997*; *Dacey et al., 2003*; *Dawis et al., 1984*; *Derrington and Lennie, 1982*; *Enroth-Cugell et al., 1983*; *Masland, 2012*; *Uzzell and Chichilnisky, 2004*). For example, nonlinear spatial summation and motion encoding have been demonstrated in parasol cells but were not utilized here (*Manookin et al., 2018*; *Turner and Rieke, 2016*). In addition, pixel-wise mean squared error does not accurately reflect the perceived quality of the visual representation. More sophisticated metrics for optimization and evaluation of reconstruction should be explored (*Wang and Lu, 2002*; *Wang et al., 2004*).

By projecting neural responses into a common stimulus space, reconstruction enabled direct comparison and evaluation of the visual signals transmitted downstream. The large collection of recordings used here revealed a consistent visual representation across retinas, in spite of differences in RF mosaic structure and firing rates that make comparing the neural response itself difficult. The information contained in the retinal signal limits the information available to downstream visual areas, so the results presented here could inform studies of visual processing in the LGN, V1, and other brain structures. For example, the oriented nature of the interaction term filters supports the hypothesis that orientation selectivity in the cortex results from pairs of nearby ON and OFF RGCs (*Paik and Ringach, 2011*; *Ringach, 2007*). In addition, comparing reconstructions from different visual areas using a standard measurement — the reconstructed image — could help reveal how information about the external world is represented at various stages of the visual system.

Using reconstruction to understand the signals transmitted by neurons may be increasingly important in future efforts to read and write neural codes using brain-machine interfaces (BMIs). In the retina, certain types of blindness can be treated with implants that use electrical stimulation to activate the remaining retinal neurons (*Goetz and Palanker, 2016*). The visual messages described in the present work could be useful for inferring the perceived visual image evoked by such devices, and thus for selecting optimal electrical stimulation patterns (*Goetz and Palanker, 2016*; *Golden et al., 2019*; *Shah et al., 2019*). Reconstruction can also be used to compare the evoked visual representation with the representation produced by natural neural activity. In addition, the observation that reconstructions from different retinas and from recorded and simulated spikes are similar suggests that perfect replication of the neural code of a particular retina may not be necessary. Outside the visual system, many BMIs rely on reconstruction to read out and interpret neural activity, for example controlling prosthetic limbs using activity recorded in the motor cortex (*Lawhern et al., 2010*; *Vargas-Irwin et al., 2010*). While these studies typically focus on performing specific tasks, the present results suggest that examination of the reconstruction filters could reveal contributions of diverse cells and cell types in these modalities.

## Materials and methods

### Experimental methods
#### Multi-electrode array recordings
An ex vivo multi-electrode array preparation was used to obtain recordings from the major types of primate RGCs (*Chichilnisky and Kalmar, 2002*; *Field et al., 2010*; *Frechette et al., 2005*; *Litke et al., 2004*). Briefly, eyes were enucleated from terminally anesthetized macaques used by other researchers in accordance with institutional guidelines for the care and use of animals. Immediately after enucleation, the anterior portion of the eye and vitreous were removed in room light, and the eye cup was placed in a bicarbonate-buffered Ames' solution (Sigma, St. Louis, MO). In dim light, pieces of retina roughly 3 mm in diameter and ranging in eccentricity from 7 to 17 mm (6–12 mm temporal equivalent eccentricity; *Chichilnisky and Kalmar, 2002*) or 29-56 degrees (*Dacey and Petersen, 1992*; *Perry and Cowey, 1985*), were placed RGC side down on a planar array consisting of 512 extracellular microelectrodes covering a 1.8 mm × 0.9 mm region (roughly 4 × 8° visual field angle). In all but one preparation, the retinal pigment epithelium (RPE) was left attached to allow for photopigment regeneration and to improve tissue stability, but the choroid (up to Bruch's membrane) was removed to allow oxygenation and maintain even thickness. For the duration of the recording, the preparation was perfused with Ames' solution (30–34° C, pH 7.4) bubbled with 95% $O_2$, 5% $CO_2$. The raw voltage traces recorded on each electrode were bandpass filtered, amplified, and digitized at 20 kHz (*Litke et al., 2004*). Spikes from individual neurons were identified by

standard spike sorting techniques, and only spike trains from cells exhibiting a 1 ms refractory period were analyzed further (*Field et al., 2007*; *Litke et al., 2004*).

## Visual stimulation

The visual stimulus was produced by a 120 Hz, gamma-corrected, CRT monitor (Sony Trinitron Multi-scan E100; Sony, Tokyo, Japan), which was optically reduced and projected through the mostly-transparent array onto the retina at low photopic light levels (2000, 1800, and 800 isomerizations per second for the L, M, and S cones respectively at 50% illumination; see *Field et al., 2009*, *Field et al., 2010*). The total visual stimulus area was 3.5 by 1.75 mm, which extended well beyond the recording area.

A 30-minutespatiotemporal white noise stimulus was used to characterize RGC responses and to periodically assess recording quality (*Chichilnisky, 2001*). The stimulus was updated at either 30 or 60 Hz, and consisted of a grid of pixels (spacing ranged from 44 to 88 µm across recordings). For each update, the intensities for each of the three monitor primaries at each pixel location were chosen randomly from a binary distribution.

Natural images from the ImageNet database (*Fei-Fei et al., 2009*) were converted to grayscale values. On a scale of 0 to 1, the mean image intensity was 0.45. The natural images were displayed at either 320 × 160 pixels, with each pixel measuring 11 × 11 µm on the retina, or at 160 × 80 pixels, with each pixel measuring 22 × 22 µm on the retina. The images were displayed for 100 ms each (12 frames at 120 Hz), separated by spatially uniform gray at intensity 0.45 for 400 ms, chosen to ensure a return to the average firing rates. The images were displayed in blocks of 1000, interleaved with a repeated set of 150 test images. Stimulation durations ranged from 5 to 40 blocks.

Dynamic movies consisted of the same set of images, each displayed for 500 ms with eye movements simulated as Brownian motion with a diffusion constant of 10 µm$^2$/frame, selected to roughly match recorded eye movements from humans (*Kuang et al., 2012*; *Van Der Linde et al., 2009*) and primate (Z.M. Hafed and R.J. Krauzlis, personal communication, June 2008). After 500 ms, a new image appeared, with no gray screen between image presentations, and again was jittered. Each recording consisted of 5000 images, for a total of 300,000 frames of stimulation.

## Cell type classification

The spike triggered average (STA) stimulus for each neuron was computed from the response to the white noise stimulus (*Chichilnisky, 2001*), to reveal the spatial, temporal, and chromatic properties of the light response. Cell type identification was performed by identifying distinct clusters in the response properties, including features of the time course and the spike train autocorrelation function extracted via principal components analysis, and the spatial extent of the RF (*Chichilnisky and Kalmar, 2002*; *Dacey, 1993*; *Devries and Baylor, 1997*; *Field et al., 2007*; *Frechette et al., 2005*). This analysis revealed multiple identifiable and complete cell type populations. In particular, the four major types, ON and OFF parasol and midget cells, were readily identifiable by their temporal properties, RF size, density, and mosaic organization (see *Rhoades et al., 2019* for a more detailed discussion). Recorded populations of parasol cells formed nearly complete mosaics over the region of retina recorded; recorded midget cell populations were less complete.

## Linear reconstruction

### Linear regression

Reconstruction filters were fitted using linear regression, as described in Results. The responses of every RGC were included in the regression for every pixel; restricting the filters to a local area did not improve reconstructions. Note that the weights for each pixel are independent, and can be fitted together or separately. Prior to regression, the distribution of each cell's responses and the pixel values at each location were centered around 0 (i.e. the mean over samples was subtracted in each case). The length of time over which spikes were counted after the image onset was chosen to optimize reconstruction performance (tested in 10 ms intervals from 10 ms to 200 ms; see *Figure 7E*). For the spike latency comparison, a maximum time of 150 ms was assigned to cells that had not yet spiked.

## Convergence of estimates

For all recordings, reconstruction performance obtained with half of the data was typically 95-98% of the reconstruction performance obtained with the full data (*Figure 12*). Both an L2-penalty on filter coefficients and applying a singular value cutoff when calculating the pseudoinverse of the response matrix (*Golden et al., 2019*; *Strang, 1980*) were tested as methods for optimizing performance with limited data. However, neither improved reconstruction performance. Note that despite the large size of the weight matrix, the appropriate comparison for fitting is samples per pixel

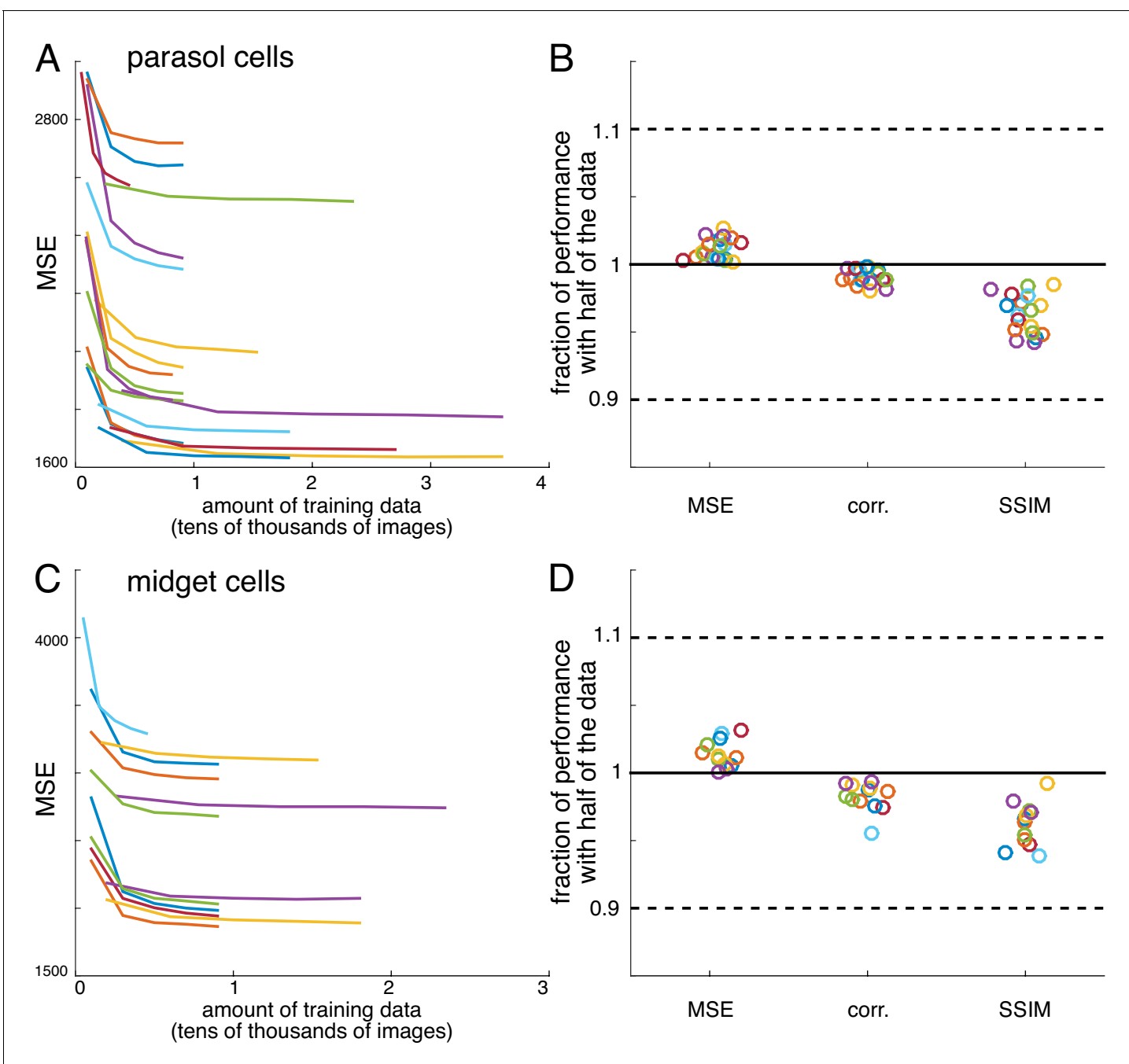

**Figure 12.** Verification of data sufficiency. (**A**) Performance of reconstructions from parasol cell responses as a function of the amount of training data, for 19 recordings (colors). (**B**) Fraction of performance of reconstructions from parasol cell responses (MSE, correlation, and SSIM) achieved with half of the training data for each recording. (**C,D**) Same as A, B for reconstructions from midget cell responses for 12 recordings (colors).

compared to weights per pixel, which is at least 20 times in every case, even when interaction terms are considered (*Figure 9*).

## Image region selection

Reconstruction performance was calculated over the image regions covered by the RFs of the recorded RGCs. To define this area, the spatial profile of each RF was fitted with a two-dimensional elliptical Gaussian (*Chichilnisky and Kalmar, 2002*), and any pixel within two standard deviations was considered covered (*Figure 13*). For each analysis, pixels were only included if they were covered by at least one of each cell type used in that analysis, so the regions included were limited by the cell type with the least coverage, typically ON or OFF midget cells. Two analyses used a manually selected, rectangular central image region instead of the mosaic coverage logic above: the comparison across recordings (*Figure 2*), and the spatial frequency analyses (*Figures 4* and *6*).

## Error metrics

The primary measures of reconstruction performance, mean squared error (MSE) and the correlation coefficient, were calculated between the original and reconstructed image, across all included pixels (as defined above). Note that linear least squares regression, which was used to obtain the filters, by definition minimizes MSE on the training data, but does not necessarily maximize the correlation coefficient. In addition, an alternative measure more closely related to perceptual difference between images, the structural similarity (SSIM; *Wang et al., 2004*), was calculated across the whole image (parameters: radius = 22 μm, exponents = [1 1 1]), and then averaged across the included pixels (see above) for each image. In all cases, similar trends were observed with each metric.

## Statistical analysis

Statistical significance was determined using resampling. In all cases presented here, two distributions of paired values were being compared, such as reconstruction performance scores for two conditions on the same set of images. To generate values in the null distribution, each pair of values was randomly distributed between the two conditions, and the mean difference was calculated. 1000 random samples were generated this way, and the p-value was the proportion of samples where the magnitude of the mean difference was greater than the recorded value. A report of $p<0.001$ indicates that no samples had a larger mean difference.

## Filter analysis

### Spatial receptive field

The spatial RF (used in *Figures 3* and *4*) was extracted from the full spatial, temporal, and chromatic spike-triggered average (STA; used for cell type classification as described above) as follows. First, the values at each pixel location and time in the STA were summed across the color channels. Significant pixels were identified as those with an absolute maximum value (across time) of more than five times the robust standard deviation of all the pixels in the STA (*Freeman et al., 2015*). Averaging across these significant pixels resulted in a single time course. The inner product of this time course with the time course of each pixel in the STA was then computed, resulting in a spatial RF.

### Average filter calculations

Average RFs (*Figure 3*) were calculated by first upsampling the spatial RFs (with linear interpolation) to match the resolution of the reconstruction filters (across recordings, scaling ranged from 2-8x), then aligning the RF centers (obtained by fitting a 2D Gaussian to the RF as described above) and averaging. Average reconstruction filters (*Figure 3*) were not upsampled, but otherwise were calculated the same way. The average RFs and filters shown in *Figure 3C* were calculated separately for each recording, cell type, and condition. A one-dimensional profile through the center of each average reconstruction filter was used to calculate full width at half maximum (*Figure 3D,E*). This calculation was robust to the angle of the profile. The average filters in *Figure 5* only included cells in regions with locally dense populations of all four major cell types (defined by the number of nearby cells of each type).

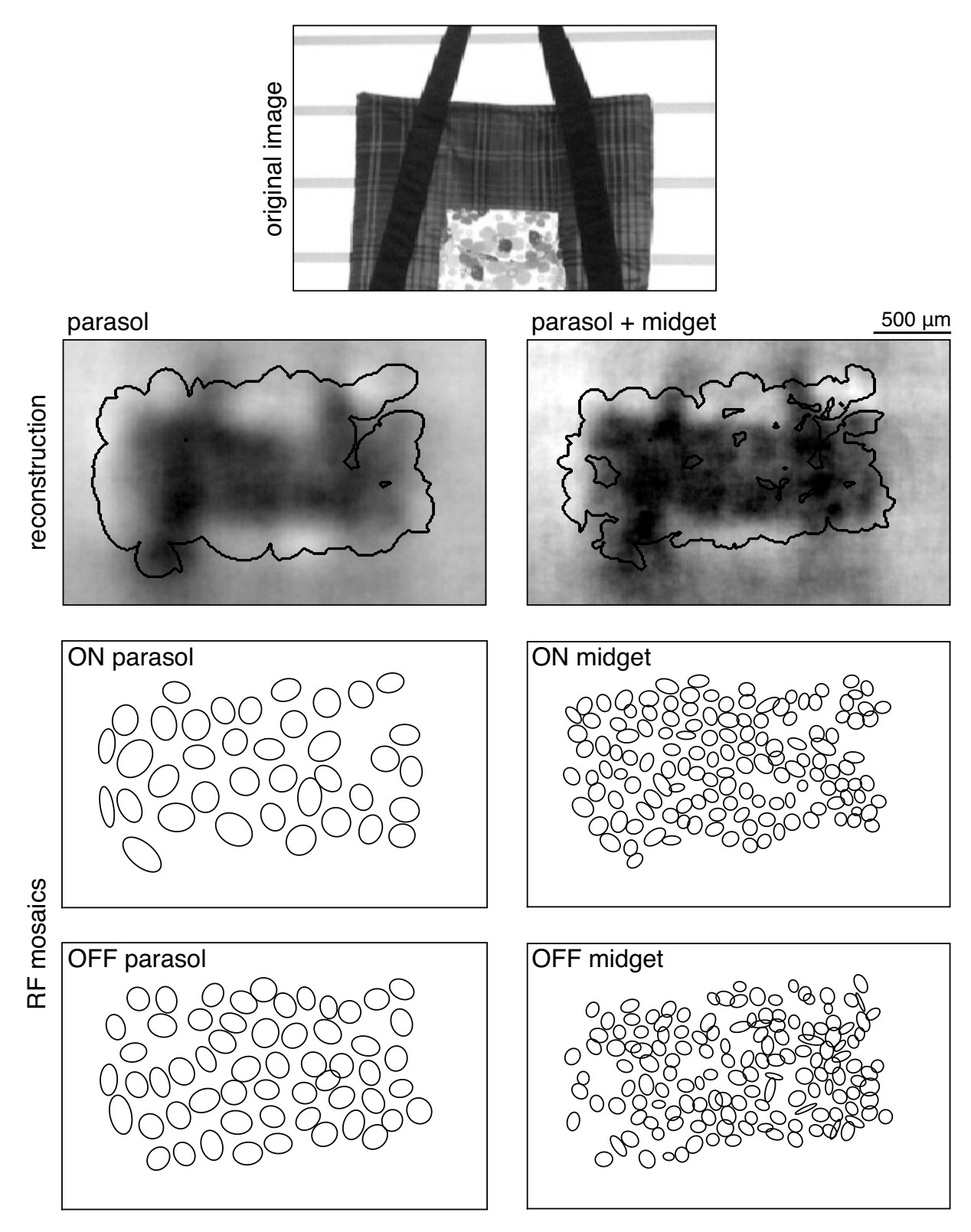

**Figure 13.** Selection of analysis region. Reconstruction performance on a sample image (top) is measured by comparing the regions inside the contours shown on the reconstructions in the second row. These contours were obtained using the receptive field mosaics (bottom two rows) of parasol cells, or of both parasol and midget cells, as described in *Image region selection*. Here, OFF midget cells had the least complete mosaic, so the included region was most limited by their coverage. The bounding boxes mark the extent of the visual stimulus.

## Receptive field reconstruction

Reconstruction from RFs (*Figure 4*) was performed as follows. Each image was estimated as a sum of RFs, weighted by the RGC response and a fitted scale factor. These scale factors were calculated by minimizing the MSE between the true and estimated images as follows:

$$a^* = argmin_a \sum_{i=1}^{n_{images}} \left(\hat{S}_i - S_i\right)^2; \; \hat{S}_i = \sum_{c=1}^{n_{cells}} F_c \cdot R_{i,c} \cdot a_c \tag{2}$$

where S is the stimulus, R is the response, F is the RF, and a is the scale factor, calculated using linear least squares regression (as described above). In this case, each pixel in each image was considered a separate sample, and was modeled as a linear combination of the image responses of all RGCs multiplied by the respective values of their RFs at that pixel. Therefore, the outputs were a vector with length equal to N, the number of images times the number of pixels in each image. The input (regressor) matrix had dimensions (N x number of cells), and the weight vector a had dimensions (number of cells x 1). For these analyses, recordings with incomplete mosaics and without high-resolution RF mapping were excluded.

## Analysis of cell type contributions

### ON and OFF parasol cells

Images were reconstructed from the responses of either ON or OFF parasol cells and performance was calculated, as described above. The relationship between true and reconstructed pixel value (*Figure 6D*) was calculated for each recording by first binning the true pixel values by percentile, resulting in bins with equal numbers of samples. Then, for each bin, the average true pixel value and the average of the corresponding reconstructed pixel values were calculated. The sensitivity was defined as the change in average reconstructed pixel value divided by the change in true pixel value across bins. The observed trends were not dependent on the number of bins.

### Parasol and midget cell classes

Images were reconstructed from the responses of either parasol or midget cell classes (including both ON and OFF types) and performance was calculated, as described above. The power spectra for the reconstructed images, original images, and average RFs (*Figure 7D*) were calculated by discrete Fourier transform. The temporal properties of the parasol and midget classes (*Figure 7E*) were compared by gradually increasing the length of the window over which spikes were counted after image onset, from 10 ms to 150 ms (in 10 ms increments). For each window size, the reconstruction filters were refitted, and the performance was calculated as described above.

For these analyses, only the recordings with the highest midget cell coverage were used, defined by the fraction of pixels included in a parasol cell analysis that would also be included in a midget cell analysis (see Image region selection above). Seven recordings were included for measuring reconstruction performance (*Figure 7C*) and comparing temporal properties (*Figure 7E*). Only three of those were also included in the spatial frequency analysis (*Figure 7D*), which required complete or nearly complete mosaics.

## Analysis of noise correlations

Noise correlation analysis (*Figure 8*) was limited to the three recordings with the most repeated presentations of the same set of test images (27 repeats each). For each of the three scenarios described in Results, reconstruction filters were fitted on a single repeat of training data, and then tested using either shuffled or unshuffled testing data. The testing data was shuffled by randomly permuting each RGC's responses independently across repeated presentations of the same image. Reconstruction performance on the test data was measured as described earlier.

## Interaction terms

Only the three recordings with the most training data were included (at least 25,000 training images each; the same subset was used for the noise correlation analysis), so that despite the increase in parameter count (from ~200 to ~1000), there were still more than enough samples to calculate the weights, and regularization did not improve cross-validated reconstruction performance.

## Linear-nonlinear simulation

Simple linear-nonlinear encoding models (*Chichilnisky, 2001*) were used to simulate spike trains for reconstruction, for each RGC independently. For each image, the inner product was first computed between the image and the spatial RF (see section Spatial receptive field above), restricted to a local region (+/- 440 μm from the RF center, corresponding to either 40 × 40 or 80 × 80 pixels depending on the resolution of the images). The resulting value was then passed through a sigmoidal nonlinearity, given by

$$y = b_4 + \frac{b_1}{b_2 + \exp(b_3 \cdot x_1)} \tag{3}$$

where the parameters $\{b_i\}$ were fitted by minimizing the mean-squared error between the predicted and measured RGC responses, on the same data set used to fit the reconstruction filters. This model was then used to simulate responses to the images used to obtain the fitting data and the images used to obtain the held-out, repeated test data. Reconstruction filters, reconstructed images, and performance were then calculated from the simulated responses in the same way as described above for the recorded responses.

## Spatiotemporal reconstruction

Each frame of the spatiotemporal movie was reconstructed using the RGC spikes recorded during that frame and the following frames. Therefore, each RGC included in the reconstruction was fitted with a full-rank, spatiotemporal reconstruction filter. The spikes were binned at the frame rate of the movie, and a filter length of 15 frames (125 ms) was selected to optimize performance. A spatial summary of the spatiotemporal filter (*Figure 11A,B*) was calculated as described above for spatial RFs. The spacetime separability of the filters was calculated using the explained variance from the first component of a singular value decomposition (limited to a spatially local region to reduce the effects of the many low-magnitude, noisy pixels outside the primary filter peak). Three recordings that contained responses to both static, flashed natural images and dynamic, spatiotemporal natural movies were included. 2400 consecutive movie frames were withheld from fitting for comparison of movie frame and static image reconstructions (*Figure 11C*).

## Acknowledgements

This work was supported by NSF IGERT 0801700 (NB), NSF GRFP DGE-114747 (NB, CR), NEI F31EY027166 (CR), Pew Charitable Trusts Fellowship in Biomedical Sciences (AS), donation from John Chen (AML), NIH R01EY017992, NIH NEI R01-EY029247, NSF/NIH CRCNS Grant IIS1430348 (EJC), and the Wu Tsai Neurosciences Institute. We thank Fred Rieke for helpful suggestions on the manuscript; Jill Desnoyer and Ryan Samarakoon for technical assistance; Sasi Madugula, Eric Wu and Alex Gogliettino for discussions and feedback; and Corinna Darian Smith and Tirin Moore (Stanford), Jose Carmena and Jack Gallant (UC Berkeley), Jonathan Horton (UCSF), and the UC Davis Primate Center for access to primate retinas.

## Additional information

### Funding

| Funder | Grant reference number | Author |
|---|---|---|
| National Science Foundation | NSF IGERT 0801700 | Nora Brackbill |
| National Science Foundation | GRFP DGE-114747 | Nora Brackbill<br>Colleen Rhoades |
| National Eye Institute | F31EY027166 | Colleen Rhoades |
| Pew Charitable Trusts | Fellowship in Biomedical Sciences | Alexander Sher |
| John Chen | Donation | Alan M Litke |
| National Institutes of Health | R01EY017992 | EJ Chichilnisky |

| National Institutes of Health | R01-EY029247 | EJ Chichilnisky |
| National Eye Institute | R01-EY029247 | EJ Chichilnisky |
| National Institutes of Health | CRCNS Grant IIS-1430348 | EJ Chichilnisky |
| National Science Foundation | CRCNS Grant IIS-1430348 | EJ Chichilnisky |
| Wu Tsai Neurosciences Institute | | EJ Chichilnisky |

The funders had no role in study design, data collection and interpretation, or the decision to submit the work for publication.

### Author contributions
Nora Brackbill, Conceptualization, Data curation, Software, Formal analysis, Funding acquisition, Validation, Investigation, Visualization, Methodology, Writing - original draft, Project administration, Writing - review and editing; Colleen Rhoades, Funding acquisition, Investigation, Writing - review and editing; Alexandra Kling, Investigation; Nishal P Shah, Investigation, Writing - review and editing; Alexander Sher, Alan M Litke, Resources, Software, Funding acquisition; EJ Chichilnisky, Conceptualization, Resources, Software, Supervision, Funding acquisition, Methodology, Project administration, Writing - review and editing

### Author ORCIDs
Nora Brackbill (ID) https://orcid.org/0000-0002-0308-1382
Nishal P Shah (ID) https://orcid.org/0000-0002-1275-0381
Alan M Litke (ID) http://orcid.org/0000-0003-3973-3642
EJ Chichilnisky (ID) https://orcid.org/0000-0002-5613-0248

### Ethics
Animal experimentation: Eyes were removed from terminally anesthetized macaque monkeys (Macaca mulatta, Macaca fascicularis) used by other laboratories in the course of their experiments, in accordance with the Institutional Animal Care and Use Committee guidelines. All of the animals were handled according to approved institutional animal care and use committee (IACUC) protocols (#28860) of the Stanford University. The protocol was approved by the Administrative Panel on Laboratory Animal Care of the Stanford University (Assurance Number: A3213-01).

### Decision letter and Author response
Decision letter https://doi.org/10.7554/eLife.58516.sa1
Author response https://doi.org/10.7554/eLife.58516.sa2

## Additional files

### Supplementary files
• Transparent reporting form

### Data availability
Code and data to generate all of the summary plots are included in the supporting files. We are not able to release the raw voltage recordings, which total >5 TBs and require a complex processing pipeline.

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
