## [Decision Letter]

**Acceptance summary:**

This article reveals "what the monkey's eye tells the monkey's brain". The authors show how one can reconstruct the visual image on the retina based on the spike signals from optic nerve fibers. Taking advantage of recordings from nearly complete populations of retinal neurons, they explore how the different types of retinal ganglion cell interact in shaping the visual message sent to the brain. The resulting rules are pleasingly simple, which may well be a design principle for the retinal code.

**Decision letter after peer review:**

Thank you for submitting your article "Reconstruction of natural images from responses of primate retinal ganglion cells" for consideration by *eLife*. Your article has been reviewed by three peer reviewers, and the evaluation has been overseen by a Reviewing Editor and Joshua Gold as the Senior Editor. The following individual involved in review of your submission has agreed to reveal their identity: Hiroki Asari (Reviewer #2).

The reviewers have discussed the reviews with one another and the Reviewing Editor has drafted this decision to help you prepare a revised submission.

We would like to draw your attention to changes in our revision policy that we have made in response to COVID-19 (https://elifesciences.org/articles/57162). Specifically, when editors judge that a submitted work as a whole belongs in *eLife* but that some conclusions require a modest amount of additional data or analysis, as they do with your paper, we are asking that the manuscript be revised to either limit claims to those supported by data in hand, or to explicitly state that the relevant conclusions require additional supporting data.

Summary:

This paper regards visual signaling by the macaque retina, specifically as viewed from the perspective of visual centers in the brain. The question is how one should interpret spike trains from retinal ganglion cells in order to reconstruct the visual image shown to the animal. Based on the established method of linear reconstruction, the authors explore how the reconstruction quality and the cells' reconstruction filters depend on the types and numbers of cells used for the reconstruction. They further study how noise correlations, nonlinear response transformations, and interactions between cells may contribute to the reconstruction. The presentation is very clear and pleasantly easy to follow, despite the technical material. The results have implications for an understanding of neural processing in the retina, and perhaps more so for the design of future retinal prostheses. In this regard the study draws particular value from using the macaque retina, which is close in function to our own.

Essential revisions:

At the same time this work comes on the background of a well-developed "standard model" of the retina that spells out how retinal ganglion cells encode the visual scene with spikes. Much of what is described here is fully expected based on that standard model. Some of the specific analyses have been done before in other species. Most of the comparison between optimal filters and single-cell receptive fields can be understood based on purely linear processing. A few other items require Linear-Nonlinear processing, which is also part of the standard model. Much of the report reads like elegant and rigorous confirmation of the conventional picture. The authors should take a clear position on the relation of their findings to the background knowledge. The reviewers can envision two possible outcomes:

A) We tried everything, but actually the optimal decoding filters are just as expected from a simple LN picture of retinal encoding. Even the noise correlations that we reported on before don't make any difference. This is good news for creating retinal prostheses because one doesn't have to engage in any sophisticated encoding.

B) We tried everything and found some interesting deviations from the conventional model of retinal function. Here they are specifically, along with the magnitude of their contributions. We expect that these deviations will have to be emulated by retinal prostheses.

In our reading, outcome (A) seems more likely, but either way the authors should choose a position.

Some specific impressions, organized by the claims in the Abstract:

1) "Each cell's visual message, defined by the optimal reconstruction filter, reflected natural image statistics, and resembled the receptive field only when nearby, same-type cells were included" This is as observed previously when reconstructing natural signals that include strong correlations. Much of the effect can be explained based on linear processing.

2) "Each cell type revealed different and largely independent visual representations, consistent with their distinct properties." This is similar to previous observations on RGCs. Independence of On and Off representations is largely explained by the opposite rectification in On and Off cells. But note midgets and parasols are not "largely independent" by this criterion. In fact the midget filter is *more* affected by including parasols than by including other midgets (Figure 5B). That is a deviation from the conventional idea of independent channels. On the other hand, one would predict these effects on the filters from the previously reported spike correlations among these types (Greschner, 2011).

3) "Stimulus-independent correlations primarily affected reconstructions from noisy responses." These noisy responses were created artificially by ignoring most of the spikes. That is not a relevant condition for actual vision. It would have been more interesting to experiment at low light levels, where prior work has shown the importance of noise. The present results seem to say that at high light levels the retinal noise is not limiting for reconstruction.

4) "Nonlinear response transformation slightly improved reconstructions with either ON or OFF parasol cells, but not both." Again the small effects seen here for On or Off cells alone are expected from the LN model. But because one has to include both On and Off cells anyway just to get the basic reconstruction correct, one can conclude that overall the nonlinear transformations don't matter for reconstruction.

5) "Inclusion of ON-OFF interactions enhanced reconstruction by emphasizing oriented edges, consistent with linear-nonlinear encoding models." These are the tiniest effects in the whole report: Δρ=0.009±0.023. The mean effect is just half of what was called "slightly improved" in point (4). It is also half of the standard deviation. Often the effect is negative, i.e. the reconstruction is worse even though the model has many more parameters; this hints at overfitting. Figure 9G is not impressive.

6) "Spatiotemporal reconstructions revealed similar spatial visual messages." A useful but "limited test" of generalization to real vision. One claim is that the spatio-temporal filters had "high space-time separability". But this leaves on the table 22% of explainable variance in the filter, which is >20 times the effect size in point (5) that got covered at great length. If, in fact, it turned out that space-time separable filters are just fine for reconstruction of videos, that would be an interesting departure from conventional wisdom, where different time course of RF center and surround have figured prominently since the 1960s.

Detailed suggestions:

7) To test which properties of retinal encoding contribute to the reconstruction filters, the authors could replace the experimental data with simulated spike trains. They have done this before to great effect using spiking LN or GLM neurons. Is reconstruction performance the same? If not, what are the important differences? Are certain aspects of visual scenes reconstructed better or worse by real cells than model cells?

8) Parasol cells are very spatially nonlinear in their responses to natural scenes (Turner and Rieke, 2016), and both ON and OFF parasols are highly motion sensitive (Manookin et al., 2018). So their use for a linear reconstruction of a static scene requires justification. Is it possible that their true role will become more apparent when reconstructing movement within the visual scene (Frechette et al., 2005)?

9) Related to 8: the conclusion that different RGC types "conveyed different and largely independent features of the visual scene" might be inappropriate when comparing midget and parasol cells. Apart from a slight difference at higher spatial frequencies, the reconstructed features of the two cell types actually seem quite similar (Figure 7).

10) More regarding the interaction between parasol and midget signals: Did adding parasol responses to midget responses aid in reconstruction simply because parasols filled in gaps in the midget mosaics? To test this, perhaps one could artificially fill in the midget mosaics with LN model cells.

11) Quality of midget vs. parasol reconstructions: RGC density is a dominant factor in image reconstruction (Results), and images reconstructed from midgets cover a wider spatial-frequency range than parasols (Figure 7D; Results). However, the reconstruction from denser midget cells is worse than that from sparser parasol cells (Figure 7C; Results). Why?

12) For the analysis that included a nonlinear, logarithmic transformation of responses, was the transformation taken into account when re-computing the weight matrix W. Also, what happened with the logarithm for bins with R=0?

13) Statistical tests: For key claims, authors should perform statistical tests to clarify the significance and better interpret the d(rho): e.g., smoothed vs. non-smoothed images, spike-counts vs. latency coding, etc.

14) Figure 13: Unclear how the reconstruction boundary relates to the RF mosaics. 'Parasol+midget' looks most like the RFs for OFF midget alone.

15) All figures: Scale bars, e.g. in degrees of visual angle, would be useful at least in key places. Also were the stimuli sized for a particular viewing distance?

16) Figure 4C: How many data points?

17) Figure 7D: What are the 3 different curves for each set? Also maybe show the 'parasol+midget' result in black?

---

## [Author Response]

Essential revisions:At the same time this work comes on the background of a well-developed "standard model" of the retina that spells out how retinal ganglion cells encode the visual scene with spikes. Much of what is described here is fully expected based on that standard model. Some of the specific analyses have been done before in other species. Most of the comparison between optimal filters and single-cell receptive fields can be understood based on purely linear processing. A few other items require Linear-Nonlinear processing, which is also part of the standard model. Much of the report reads like elegant and rigorous confirmation of the conventional picture. The authors should take a clear position on the relation of their findings to the background knowledge. The reviewers can envision two possible outcomes:A) We tried everything, but actually the optimal decoding filters are just as expected from a simple LN picture of retinal encoding. Even the noise correlations that we reported on before don't make any difference. This is good news for creating retinal prostheses because one doesn't have to engage in any sophisticated encoding.B) We tried everything and found some interesting deviations from the conventional model of retinal function. Here they are specifically, along with the magnitude of their contributions. We expect that these deviations will have to be emulated by retinal prostheses.In our reading, outcome (A) seems more likely, but either way the authors should choose a position.

We thank the reviewers for the opportunity to refine the main message of the paper, and we agree that the findings primarily support interpretation (A). As suggested, we have compared the results to simulations using a simple linear-nonlinear (LN) model of RGC light response. We found a strong correspondence in both the reconstruction filters and the reconstructed images, which provides additional support for interpretation (A). Details of this comparison are described further under point (7), and these results are described in the manuscript in the newly added subsection “Comparison to simple models of RGC light response”. This section should also help clarify the small impacts of noise correlations and nonlinear reconstruction (mentioned by the reviewers below under points 4 and 5).

Some specific impressions, organized by the claims in the Abstract:1) "Each cell's visual message, defined by the optimal reconstruction filter, reflected natural image statistics, and resembled the receptive field only when nearby, same-type cells were included" This is as observed previously when reconstructing natural signals that include strong correlations. Much of the effect can be explained based on linear processing.

Yes, we found that the optimal reconstruction filters were consistent with what would be expected from simple encoding models that have the same spatial structure and organization as the measured RFs, a key aspect of the data presented here. A comparison using simulated spike trains is described below, under point (7).

2) "Each cell type revealed different and largely independent visual representations, consistent with their distinct properties." This is similar to previous observations on RGCs. Independence of On and Off representations is largely explained by the opposite rectification in On and Off cells. But note midgets and parasols are not "largely independent" by this criterion. In fact the midget filter is more affected by including parasols than by including other midgets (Figure 5B). That is a deviation from the conventional idea of independent channels. On the other hand, one would predict these effects on the filters from the previously reported spike correlations among these types (Greschner, 2011).

Yes, we agree that this analysis revealed that the spatial information carried by the parasol and midget cell classes was not quite independent, and we have updated the manuscript to emphasize this point (Abstract, Introduction, Results, Discussion). The asymmetric nature of the effect (parasols affect midgets, but not vice-versa) is also interesting, and likely reflects differences in response strength (see the response to point (11) below). It also seems likely that future investigations of temporal encoding will clarify the independence of these channels, a point we have now mentioned more prominently in the manuscript (Discussion).

3) "Stimulus-independent correlations primarily affected reconstructions from noisy responses." These noisy responses were created artificially by ignoring most of the spikes. That is not a relevant condition for actual vision. It would have been more interesting to experiment at low light levels, where prior work has shown the importance of noise. The present results seem to say that at high light levels the retinal noise is not limiting for reconstruction.

We assumed that using a short time window, which captured the peak firing activity but did not include every stimulus driven spike, *does* relate to natural vision: some important aspects of perception happen very rapidly rather than awaiting the arrival of all of the stimulus-driven spikes from the retina. However, the reviewer is right that we assumed that this approach would be a way to emulate a low-fidelity encoding situation such as occurs at low light levels, without performing an empirical test. We have edited the text to clarify these assumptions (subsection “The effect of correlated firing”).

4) "Nonlinear response transformation slightly improved reconstructions with either ON or OFF parasol cells, but not both." Again the small effects seen here for On or Off cells alone are expected from the LN model. But because one has to include both On and Off cells anyway just to get the basic reconstruction correct, one can conclude that overall the nonlinear transformations don't matter for reconstruction.

We agree that these results support the conclusion that nonlinear transformations are not important for subsequent linear reconstruction when both ON and OFF types are included, as in natural vision. However, a priori, reconstructions could have required only ON or OFF cells, if the appropriate nonlinear transformation were used prior to reconstruction. For example, the responses of the ON parasol cells could potentially contain enough information to reconstruct dark regions of the image, but not in a linear way. This would be the case if the encoding nonlinearity were not strongly rectified, and could be corrected for, and would imply that downstream targets receiving information from just ON cells could fully represent the image. However, we did not find that to be the case. We have attempted to be more explicit about this in the text, and have provided a direct comparison between reconstructions from the transformed responses of a single cell type and the original responses of both cell types (subsection “Nonlinear reconstruction”).

5) "Inclusion of ON-OFF interactions enhanced reconstruction by emphasizing oriented edges, consistent with linear-nonlinear encoding models." These are the tiniest effects in the whole report: Δρ=0.009±0.023. The mean effect is just half of what was called "slightly improved" in point (4). It is also half of the standard deviation. Often the effect is negative, i.e. the reconstruction is worse even though the model has many more parameters; this hints at overfitting. Figure 9G is not impressive.

We realize that the effect size in this analysis is quite small, which we had intended to convey. However, we still think this point needs to be thoroughly documented, because it was not obvious or known from previous studies. We have updated the text in these sections to emphasize the limited overall impact, and we hope that the new subsection “Comparison to simple models of RGC light response” further clarifies the interpretation of this analysis. In addition, even though there are more parameters, we did not find improvement using regularization, which argues against overfitting. This is likely because even with the interaction terms, each image pixel has roughly 1000 weights, which are estimated from more than 25,000 samples (these weights are not influenced by the weights for other pixels).

6) "Spatiotemporal reconstructions revealed similar spatial visual messages." A useful but "limited test" of generalization to real vision. One claim is that the spatio-temporal filters had "high space-time separability". But this leaves on the table 22% of explainable variance in the filter, which is >20 times the effect size in point (5) that got covered at great length. If, in fact, it turned out that space-time separable filters are just fine for reconstruction of videos, that would be an interesting departure from conventional wisdom, where different time course of RF center and surround have figured prominently since the 1960s.

We agree, this is only a limited test of the filters that would be relevant for natural vision, intended to identify whether there were gross differences between the spatial information present in reconstructions of images vs. movies. The remaining, unexplained variance in the spatiotemporal filters (beyond the separable approximation) is a combination of noise and structure, and the structure could in principle have a significant impact on reconstruction. Furthermore, it could have a biological interpretation, for example the center and surround timing mentioned by the reviewer. We have edited the text to point out that some inseparability would be expected and may be important, and to emphasize that generalization to real vision will require more data and analysis (subsection “Spatial information in a naturalistic movie”, Discussion, sixth paragraph).

Detailed suggestions:7) To test which properties of retinal encoding contribute to the reconstruction filters, the authors could replace the experimental data with simulated spike trains. They have done this before to great effect using spiking LN or GLM neurons. Is reconstruction performance the same? If not, what are the important differences? Are certain aspects of visual scenes reconstructed better or worse by real cells than model cells?

We thank the reviewers for this suggestion, which we have incorporated into the manuscript (see the new subsection “Comparison to simple models of RGC light response”). The resulting reconstruction filters and reconstructed images found using simulated spike trains, which were fitted for each recorded RGC and therefore incorporated measured spatial and response properties, were very similar to those found using recorded spike trains. While it is still possible that more sophisticated reconstruction models may utilize additional properties of retinal encoding, the visual message described here is therefore consistent with a simple LN encoding model. We have modified the emphasis at several places in the document to be explicit about this, as suggested.

8) Parasol cells are very spatially nonlinear in their responses to natural scenes (Turner and Rieke, 2016), and both ON and OFF parasols are highly motion sensitive (Manookin et al., 2018). So their use for a linear reconstruction of a static scene requires justification. Is it possible that their true role will become more apparent when reconstructing movement within the visual scene (Frechette et al., 2005)?

While the responses of parasol cells contained enough intensity information to enable linear reconstruction of images from their signals, they likely also contain specific information about motion and fine spatial detail, as described in the papers mentioned here. It is not clear how encoding complex features such as these relates to reconstruction, as it has been suggested that nonlinear encoding in the early visual pathways may enable simpler downstream reconstruction (DiCarlo et al., 2012; Rieke, 1997; Gjorgjieva et al., 2019; Naselaris et al., 2011). However, it is certainly possible that reconstruction of more complex, dynamic natural stimuli would reveal additional stimulus features conveyed by parasol cells (and by other RGCs). Extracting those features would probably require a more complex decoding scheme, potentially using a different loss function, an area we are currently exploring. We have elaborated on this point in the Discussion (sixth paragraph).

9) Related to 8: the conclusion that different RGC types "conveyed different and largely independent features of the visual scene" might be inappropriate when comparing midget and parasol cells. Apart from a slight difference at higher spatial frequencies, the reconstructed features of the two cell types actually seem quite similar (Figure 7).

Yes, as discussed above in point (2), we agree that this is not the whole story when comparing parasol and midget cell classes. We have updated the text to more clearly emphasize this (Abstract, Introduction, Results, Discussion). In addition, see the response to point (11) below for some additional details on the differences in reconstructed features.

10) More regarding the interaction between parasol and midget signals: Did adding parasol responses to midget responses aid in reconstruction simply because parasols filled in gaps in the midget mosaics? To test this, perhaps one could artificially fill in the midget mosaics with LN model cells.

The reconstruction performance was only measured in areas of the image that were covered by receptive fields of all cell types (i.e. including midget cells). This is described in the Materials and methods (subsection “Image region selection”) and illustrated in Figure 13 (also see the response to point (14) below). We believe this addresses the question, however, if we have missed something we could revisit this.

11) Quality of midget vs. parasol reconstructions: RGC density is a dominant factor in image reconstruction (Results), and images reconstructed from midgets cover a wider spatial-frequency range than parasols (Figure 7D; Results). However, the reconstruction from denser midget cells is worse than that from sparser parasol cells (Figure 7C; Results). Why?

While images reconstructed from midget cell responses had clearer edges and finer spatial details, they typically did not capture the overall luminance values across the image as well as parasol cells. The images reconstructed from parasol cell responses had 50% higher SNR. Together, these led to lower correlation scores for the reconstructions from midget cell responses. We have updated the text to reflect this (subsection “Distinct contributions of major cell types”).

12) For the analysis that included a nonlinear, logarithmic transformation of responses, was the transformation taken into account when re-computing the weight matrix W. Also, what happened with the logarithm for bins with R=0?

Yes, the reconstruction weights were recalculated based on the transformed responses. In addition, the log transformation included a +1 term, which we neglected to mention. The text has been updated to include these points (subsection “Nonlinear reconstruction”).

13) Statistical tests: For key claims, authors should perform statistical tests to clarify the significance and better interpret the d(rho): e.g., smoothed vs. non-smoothed images, spike-counts vs. latency coding, etc.

We have included statistical tests (computed via resampling, described in the Materials and methods) for the following analyses in the text. Due to large sample sizes, and therefore high statistical power, many of the differences were significant even when the effect sizes were small.

– Smoothed vs. non-smoothed images

– Spike-counts vs. latency coding

– RF vs. filter coverage

– STA vs. original reconstructions

– Reconstruction performance across cell types and classes (ON vs. OFF vs. both, parasol vs. midget vs. both)

– Impact of noise correlations, nonlinear transformations, and interaction terms

14) Figure 13: Unclear how the reconstruction boundary relates to the RF mosaics. 'Parasol+midget' looks most like the RFs for OFF midget alone.

The combined reconstruction boundary is the intersection of areas for each of the four cell types, to ensure that only areas with complete coverage of all four types were considered when measuring performance. In the example shown, this intersection was primarily limited by the less complete OFF midget cell mosaic. The figure legend (Figure 13) has been updated to clarify this.

15) All figures: Scale bars, e.g. in degrees of visual angle, would be useful at least in key places. Also were the stimuli sized for a particular viewing distance?

Scale bars have been included in terms of the size of the projected visual stimulus on the retina. We used millimeters, which we can precisely measure. However, a rough conversion to visual angle is included in the Materials and methods. The stimuli were not sized for a particular viewing distance, as natural scenes have structure and objects at many scales, so there is no inherently appropriate distance.

16) Figure 4C: How many data points?

Figure 4C includes 24 data points, one for each cell type (ON or OFF parasol) from each of 12 experiments (3 of the original 15 were excluded due to incomplete mosaics). This information has been added to the figure legend, and a description of the selection criteria was added to the Materials and methods.

17) Figure 7D: What are the 3 different curves for each set? Also maybe show the 'parasol+midget' result in black?

The curves are the 3 recordings used in this analysis (selection of the recordings is described in the Materials and methods). Since a smaller number of recordings were used here, the individual curves were shown rather than an average. The legend and labels have been updated to reflect this. The combined parasol and midget reconstructions were also added as suggested.